# G-PATE: Scalable Differentially Private Data Generator via Private Aggregation of Teacher Discriminators

**Yunhui Long**[1*]  **Boxin Wang**[1*]  **Zhuolin Yang**[1]  **Bhavya Kailkhura**[2]  **Aston Zhang**[1]

**Carl A. Gunter**[1]  **Bo Li**[1]

[1] University of Illinois, Urbana Champaign   [2] Lawrence Livermore National Laboratory
{ylong4, boxinw2, zhuolin5, lzhang74, cgunter, lbo}@illinois.edu

## Abstract

Recent advances in machine learning have largely benefited from the massive accessible training data. However, large-scale data sharing has raised great privacy concerns. In this work, we propose a novel privacy-preserving data Generative model based on the PATE framework (G-PATE), aiming to train a scalable differentially private data generator which preserves high generated data utility. Our approach leverages generative adversarial nets to generate data, combined with private aggregation among different discriminators to ensure strong privacy guarantees. Compared to existing approaches, G-PATE significantly improves the use of privacy budgets. In particular, we train a student data generator with an ensemble of teacher discriminators and propose a novel private gradient aggregation mechanism to ensure differential privacy on all information that flows from teacher discriminators to the student generator. In addition, with random projection and gradient discretization, the proposed gradient aggregation mechanism is able to effectively deal with high-dimensional gradient vectors. Theoretically, we prove that G-PATE ensures differential privacy for the data generator. Empirically, we demonstrate the superiority of G-PATE over prior work through extensive experiments. We show that G-PATE is the first work being able to generate high-dimensional image data with high data utility under limited privacy budgets ($\varepsilon \leq 1$). Our code is available at https://github.com/AI-secure/G-PATE.

## 1 Introduction

Machine learning has been applied to a wide range of applications such as face recognition [30, 39, 21, 22], autonomous driving [26], and medical diagnoses [8, 20]. However, most learning methods rely on the availability of large-scale training datasets containing sensitive information such as personal photos or medical records. Therefore, such sensitive datasets are often hard to be shared due to privacy concerns [40]. To handle this challenge, data providers sometimes release synthetic datasets produced by generative models learned on the original data. Though recent studies show that generative models such as generative adversarial networks (GAN) [14] can generate synthetic records that are indistinguishable from the original data distribution, there is no theoretical guarantee on the privacy protection. While privacy definitions such as differential privacy [9] and Rényi differential privacy [27] provide rigorous privacy guarantee, applying them to synthetic data generation is nontrivial.

Recently, two approaches have been proposed to combine differential privacy with synthetic data generation: DP-GAN [35] and PATE-GAN [37]. DP-GAN modifies GAN by training the discriminator using differentially private stochastic gradient descent. Though it achieves privacy guarantee due to

---

*Equal contribution.

the post processing property [10] of differential privacy, DP-GAN incurs significant utility loss on the synthetic data, especially when the privacy budget is low. In contrast, PATE-GAN trains differentially private GAN using Private Aggregation of Teacher Ensembles (PATE) [28]. Specifically, it trains a set of teacher discriminators and a student discriminator. To ensure differential privacy, the student discriminator is only trained on records that are produced by the generator and labeled by the teacher discriminators. The key limitation of this approach is that it relies on the assumption that the generator would be able to generate the entire real records space to bootstrap the training process. If most of the synthetic records are labeled as fake by the teacher discriminators, the student discriminator would be trained on a biased dataset and fail to learn the true data distribution. Consequently, this trained generator would not be able to produce high-quality synthetic data. This problem does not exist for traditional GAN, where the discriminator is always able to provide useful information to the generator since they can access the real data records rather than the synthetic data only.

The main contribution of this paper is a new approach named G-PATE for training a differentially private data generator by combining the generative model with PATE mechanism. Our approach is based on the key observation that: *It is not necessary to ensure differential privacy for the discriminator in order to train a differentially private generator*. As long as we ensure differential privacy on the information flow from the discriminator to the generator, it is sufficient to guarantee the privacy property for the generator. To achieve this, we propose a private gradient aggregation mechanism to ensure differential privacy on all the information that flows from the teacher discriminators to the student generator. The aggregation mechanism applies random projection and gradient discretization to reduce privacy budget consumed by each aggregation step and to increase model scalability. Compared to PATE-GAN, our approach has three advantages. First, it improves the use of privacy budget by only applying it to the part of the model that actually needs to be released for data generation. Second, our discriminator can be trained on original data records since it does not need to satisfy differential privacy. Finally, G-PATE preserves better utility on high-dimensional data given its more efficient gradient aggregation mechanism.

Theoretically, we show that our algorithm ensures differential privacy for the generator. Empirically, we conduct extensive experiments on the Kaggle credit dataset and image datasets. To the best of our knowledge, this is the first work that is able to scale to high-dimensional face image dataset such as CelebA while still preserving high data utility. The results show that our method significantly outperforms all baselines including DP-GAN and PATE-GAN.

## 2 Related Work

Differential privacy [9] is a notion that ensures an algorithm outputs general information about its input dataset without revealing individual information. So far, researchers have proposed different methods to design differentially private statistical functions and machine learning models [2, 5, 1, 25, 12]. Private aggregation of teacher ensembles (PATE) is proposed to train a differentially private classifier using ensemble mechanisms. Scalable PATE [29] improves the utility of PATE with a Confident-GNMax aggregator that only returns a result if it has high confidence in the consensus among teachers. However, PATE and Scalable PATE are only applicable to categorical data (i.e., class labels) as shown in prior work. Differentially private data generative models have also been proposed. Priview [31] generates synthetic data based on marginal distributions of the original dataset, PrivBayes [38] trains a differentially private Bayesian network, and MWEM [16] uses the multiplicative weights framework to maintain and improve a distribution approximating a given data set with respect to a set of counting queries. However, these approaches are not suitable for image datasets since the statistics they use cannot well preserve the correlations between pixels in an image.

Some recent work applies differential privacy to the training of GAN. DP-GAN [35] and DP-CGAN [33] add Gaussian noise to the gradients of the discriminators during the training process. GS-WGAN [6] reduces gradient sensitivity using the Wasserstein distance and uses gradient sanitization to ensure differential privacy for the generator. PATE-GAN [37] trains a student discriminator using an ensemble of teacher discriminators. DP-MERF[15] and PEARL [23] use differentially private embedding to train generative models on the embedding space. Concretely, DP-MERF uses random feature representations of kernel mean embeddings, and PEARL improves upon DP-MERF by incorporating characteristic function that improves the generator's learning capability. Both DP-MERF and PEARL focus on generating a private embedding space on which the distance metric between synthetic and real data is computed (though there are no results reported for high-dimensional images). On the contrary, G-PATE focuses on generating DP high-dimensional data by improving

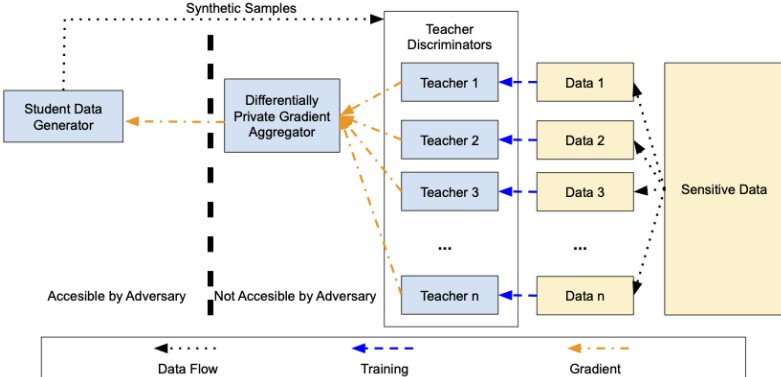

Figure 1: **Model Overview of G-PATE.** The model contains three parts: a student data generator, a differentially private gradient aggregator, and an ensemble of teacher discriminators.

the model structure and the private gradient aggregation step, which is orthogonal to the embedding space optimization approaches as DP-MERF and PEARL.

## 3  Preliminaries

**Differential Privacy.**    Differential privacy bounds the shift in the output distribution of a randomized algorithm that could be caused by a small input perturbation. The following definition formally describes this privacy guarantee.

**Definition 1** (($\varepsilon, \delta$)-Differential Privacy). A randomized algorithm $\mathcal{M}$ with domain $\mathbb{N}^{|\mathcal{X}|}$ is ($\varepsilon, \delta$)-differentially private if for all $\mathcal{S} \subseteq \text{Range}(\mathcal{M})$ and for any neighboring datasets $D$ and $D'$, we have $\Pr[\mathcal{M}(D) \in \mathcal{S}] \leq \exp(\varepsilon) \Pr[\mathcal{M}(\mathcal{D}') \in \mathcal{S}] + \delta$.

**Rényi Differential Privacy.**    Rényi differential privacy is a natural relaxation of differential privacy. Defined below, its privacy guarantee is expressed in terms of Rényi divergence.

**Definition 2** (($\lambda, \varepsilon$)-RDP). A randomized mechanism $\mathcal{M}$ is said to guarantee ($\lambda, \varepsilon$)-RDP with $\lambda > 1$ if for any neighboring datasets $D$ and $D'$,

$$D_\lambda \left( \mathcal{M}(D) \| \mathcal{M}(D') \right) =$$

$$\frac{1}{\lambda - 1} \log \mathbb{E}_{x \sim \mathcal{M}(D)} \left[ \left( \frac{\mathbf{Pr}[\mathcal{M}(D) = x]}{\mathbf{Pr}\left[\mathcal{M}(D') = x\right]} \right)^{\lambda - 1} \right] \leq \varepsilon.$$

($\lambda, \varepsilon$)-RDP implies ($\varepsilon_\delta, \delta$)-differential privacy for any given probability $\delta > 0$.

**Theorem 1** (From RDP to DP). *If a mechanism $\mathcal{M}$ guarantees ($\lambda, \varepsilon$)-RDP, then $\mathcal{M}$ guarantees $(\varepsilon + \frac{\log 1/\delta}{\lambda - 1}, \delta)$-differential privacy for any $\delta \in (0, 1)$.*

Compared to DP, RDP supports easier composition of multiple queries and clearer privacy guarantee under Gaussian noise. Specifically, RDP could be easily composed by adding the privacy budget:

**Theorem 2** (Composition of RDP). *If a mechanism $\mathcal{M}$ consists of a sequence of $\mathcal{M}_1, \ldots, \mathcal{M}_k$ such that for any $i \in [k]$, $\mathcal{M}_i$ guarantees ($\lambda, \varepsilon_i$)-RDP, then $\mathcal{M}$ guarantees $(\lambda, \sum_{i=1}^{k} \varepsilon_i)$-RDP.*

Suppose $f$ is a real-valued function, and the Gaussian mechanism is defined as $\mathbf{G}_\sigma f(D) = f(D) + \mathcal{N}\left(0, \sigma^2\right)$, where $\mathcal{N}\left(0, \sigma^2\right)$ is normally distributed random variable with standard deviation $\sigma$ and mean 0. The Gaussian mechanism provides the following RDP guarantee:

**Theorem 3** (RDP Guarantee for Gaussian Mechanism). *If $f$ has sensitivity 1, then the Gaussian mechanism $\mathbf{G}_\sigma f$ satisfies $\left(\lambda, \lambda/\left(2\sigma^2\right)\right)$-RDP.*

## 4  G-PATE: A Scalable Data Generative Method

In this section, we present our method named G-PATE. An overview of the method is shown in Figure 1. Unlike PATE-GAN and DP-GAN, G-PATE ensures differential privacy for the information

flow from the discriminator to the generator. This improvement incurs less utility loss on the synthetic samples, so it can generate synthetic samples for higher dimensional and more complex datasets.

G-PATE makes two major modifications on the training process of GAN. First, we replace the discriminator in GAN with an ensemble of teacher discriminators trained on disjoint subsets of the sensitive data. The teacher discriminators do not need to be published, thus can be trained using non-private algorithms. In addition, we design a gradient aggregator to collect information from teacher discriminators and combine them in a differentially private fashion. The output of the aggregator is a gradient vector that guides the student generator to improve its synthetic samples.

Unlike PATE-GAN, G-PATE does not require any student discriminator. The teacher discriminators are directly connected to the student generator. The gradient aggregator adds noise in the information flow from the teacher discriminators to the student generator to ensure differential privacy. This way, G-PATE uses privacy budget more efficiently and better approximates the real data distribution to ensure high data utility.

## 4.1 Training the Student Generator

To achieve better privacy budget efficiency, G-PATE only ensures differential privacy for the generator and allows the discriminators to learn private information.

To ease privacy analysis, we decompose G-PATE into three parts: the teacher discriminators, the student generator, and the gradient aggregator. To prevent the propagation of private information, the student generator does not have direct access to any information in any of the teacher discriminators. Consequently, we cannot train the student generator by ascending its gradient based on loss of the discriminators. To solve this problem, we calculate the backpropagated gradients on the fake record $x$ by ascending $x$'s gradients on the loss of the discriminator. This gradient vector can be viewed as an adversarial perturbation on $x$ that would cause the discriminator's loss on $x$ to increase. Therefore, adding the gradients to the generated fake record would teach the student generator how to improve the fake record. In each training iteration, the student generator is updated in three steps: (1) A teacher discriminator generates the backpropagated gradients for each record produced by the student generator. (2) The gradient aggregator takes the gradients from all teacher models and generates a differentially private aggregation of them. (3) The student generator updates its weights based on the privately aggregated gradients. The process is formally presented in Algorithm 1.

---

**Algorithm 1 - Training the Student Generator.**

1: **Input:** batch size $m$, number of teacher models $n$, number of training iterations $N$, gradient clipping constant $c$, number of bins $B$, projected dimension $k$, noise parameters $\sigma_1$ and $\sigma_2$, threshold $T$, disjoint subsets of sensitive data $S_1, S_2, \ldots, S_n$
2: **for** number of training iterations **do**
3:     *//Phase I: Pre-Processing*
4:     Sample $m$ noise samples $\mathbf{z_1}, \mathbf{z_2}, \ldots, \mathbf{z_m}$
5:     Generate fake samples $G(\mathbf{z_1}), G(\mathbf{z_2}), \ldots, G(\mathbf{z_m})$
6:     **for** each synthetic image $G(\mathbf{z_j})$ **do**
7:         *//Phase II: Private computation and aggregation*
8:         **for** each teacher model $i$ **do**
9:             Sample $m$ data samples from $S_i$
10:             Update the teacher discriminator $D_i$
11:             Calculate the gradients $\Delta\mathbf{x}_\mathbf{j}^{(\mathbf{i})}$
12:         **end for**
13:         $\Delta\mathbf{X_j} \leftarrow \left( \Delta\mathbf{x}_\mathbf{j}^{(\mathbf{1})}; \Delta\mathbf{x}_\mathbf{j}^{(\mathbf{2})}; \ldots; \Delta\mathbf{x}_\mathbf{j}^{(\mathbf{n})} \right)$
14:         $\Delta\mathbf{x}_\mathbf{j}^{\mathrm{priv}} \leftarrow \mathtt{DPGradAgg}\left( \Delta\mathbf{X_j}, c, B, k, \sigma_1, \sigma_2, T \right)$
15:         *//Phase III: Post-Processing*
16:         $\hat{\mathbf{x}}_\mathbf{j} \leftarrow G(\mathbf{z_j}) + \Delta\mathbf{x}_\mathbf{j}^{\mathrm{priv}}$
17:     **end for**
18:     Update the student generator $G$ by descending its stochastic gradient on $\mathcal{L}_G$ on $(\hat{\mathbf{x}}_\mathbf{1}, \hat{\mathbf{x}}_\mathbf{2}, \ldots, \hat{\mathbf{x}}_\mathbf{m})$
19: **end for**

---

**Backpropagating Gradients in the Discriminator.** Let $D$ be a teacher discriminator. Given a fake record $x$, we use $\mathcal{L}_D(x)$ to represent $D$'s loss on $x$. In each training iteration, the weights of $D$ are updated by descending their stochastic gradients on $\mathcal{L}_D$.

For each input fake record $x$, we generate a gradient vector $\Delta x$ that guides the student generator on improving its output. By applying the perturbation on its output, the student generator would get an improved fake record $\hat{x} = x + \Delta x$ on which $D$ has a higher loss. Therefore, $\Delta x$ is calculated as $x$'s gradients on $\mathcal{L}_D$:

$$\Delta x = \left. \frac{\partial \mathcal{L}_D(a)}{\partial a} \right|_{a=x}. \tag{1}$$

With the gradient vector $\Delta x$, the student generator can be trained without direct access to the discriminator's loss.

**Updating the Student Generator.** A student generator $G$ learns to map a random input $z$ to a fake record $x = G(z)$ so that $x$ is indistinguishable from a real record by $D$. Given the gradient vector $\Delta x$, the teacher discriminators have higher loss on the perturbed fake record $\hat{x} = x + \Delta x$ compared to the original fake record $x$. Therefore, the student generator learns to improve its fake records by minimizing the mean squared error (MSE) between its output $G(z)$ and the perturbed fake record $\hat{x}$.

$$\mathcal{L}_G(z, \hat{x}) = \frac{1}{k} \sum_{i=1}^{k} (G(z_i) - \hat{x}_i)^2, \tag{2}$$

where $k$ is the number of synthetic records generated per training iteration. To ensure differential privacy, instead of receiving the gradient vector from a single discriminator, we train the student generator using a differentially private gradient aggregator that combines gradient vectors from multiple teacher discriminators. Details are provided in Section 4.2.

## 4.2 Differentially Private Gradient Aggregation for G-PATE

G-PATE consists of a student generator and an ensemble of teacher discriminators trained on disjoint subsets of the sensitive data. In each training iteration, each teacher discriminator generates a gradient vector $\Delta x$ that guides the student generator on improving its output records. Different from traditional GAN, in G-PATE, the student generator does not have access to the loss of any teacher discriminators, and the gradient vector is the only information propagated from the teacher discriminators to the student generator. Therefore, to achieve differential privacy, it suffices to add noise during the aggregation of the gradient vectors.

However, the aggregators used in PATE and PATE-GAN are not suitable for aggregating gradient vectors because they are only applicable to categorical data. Therefore, we propose a differentially private gradient aggregator (DPGradAgg) based on PATE. With gradi-

---

**Algorithm 2 - Differentially Private Gradient Aggregator (DPGradAgg).** This algorithm takes a list of gradient vectors and returns a differentially private aggregation.

1: **Input:** Concatenated gradient vectors from each teacher model $\Delta \mathbf{X} = (\Delta \mathbf{x}^{(1)}, \ldots, \Delta \mathbf{x}^{(n)})$, gradient clipping constant $c$, number of bins $B$, projected dimension $k$, noise parameters $\sigma_1$ and $\sigma_2$, threshold $T$
2: $k_0 \leftarrow$ the dimension of $\Delta \mathbf{x}^{(1)}$
3: $\mathbf{R} \leftarrow$ a $k_0 \times k$ random projection matrix with each component randomly drawn from $\mathcal{N}(0, \frac{1}{k})$
4: $\Delta \mathbf{U} \leftarrow \Delta \mathbf{X} \mathbf{R}$
5: **for** each column $\mathbf{u_j}$ of $\Delta \mathbf{U}$ **do**
6:     Clip $\mathbf{u_j}$ to $(-c, c)$
7:     $h \leftarrow$ the histogram of $\mathbf{u_j}$ with $B$ bins of width $\frac{2c}{B}$
8:     $\Delta u_j^{\text{priv}} \leftarrow \text{Confident-GNMax}(h, \sigma_1, \sigma_2, T)$
9: **end for**
10: $\Delta \mathbf{u}^{\text{priv}} \leftarrow (\Delta u_1^{\text{priv}}; \ldots; \Delta u_k^{\text{priv}})$
11: $\Delta \mathbf{x}^{\text{priv}} \leftarrow \Delta \mathbf{u}^{\text{priv}} \mathbf{R}^{\mathsf{T}}$
12: **Output:** $\Delta \mathbf{x}^{\text{priv}}$

---

ent discretization, we convert gradient aggregation into a voting problem and get the noisy aggregation of teachers' votes using PATE. Additionally, we use random projection to reduce the dimension of vectors on which the aggregation is performed. The combination of these two approaches allows G-PATE to generate synthetic samples with higher data utility, even for large scale image datasets, which is hard to be achieved by PATE-GAN. The procedure is formally presented in Algorithm 2.

**Gradient Discretization.** Since PATE is originally designed for aggregating the teacher models' votes on the correct class label of an example, the aggregation mechanism in PATE only applies to categorical data. Therefore, we design a three-step algorithm to apply PATE on continuous gradient vectors. First, we discretize the gradient vector by creating a histogram and mapping each element to the midpoint of the bin it belongs to. Then, instead of voting for the class labels as in PATE, a teacher discriminator votes for $k$ bins associated with $k$ elements in its gradient vector. Finally, for each dimension, we calculate the bin with most votes using the Confident-GNMax aggregator [29] (Appendix D). The aggregated gradient vector consists of the midpoints of the selected bins.

With gradient discretization, the teacher discriminators can directly communicate with the student generator using the PATE mechanism. Since these teacher discriminators are trained on real data, they can provide much better guidance to the generator compared to the student discriminator in PATE-GAN, which is only trained on synthetic samples. Moreover, the Confident-GNMax aggregator ensures that the student generator would only improve its output in the direction agreed by most of the teacher discriminators.

**Random Projection.** Aggregation of high dimensional vectors is expensive in terms of privacy budget because private voting needs to be performed on each dimension of the vectors. To save

privacy budget, we use random projection [3] to reduce the dimensionality of gradient vectors. Before the aggregation, we generate a random projection matrix with each component randomly drawn from a Gaussian distribution. We then project the gradient vector into a lower dimensional space using the random projection matrix. After the aggregation, the aggregated gradient vector is projected back to its original dimensions. Since the generation of random projection matrix is data-independent. It does not consume any privacy budgets.

Random projection is shown to be especially effective on image datasets. Since different pixels of an image are often highly correlated, the intrinsic dimension of an image is usually much lower than the number of pixels [13]. Therefore, random projection maximizes the amount of information a student generator can get from a single query to the `Confident-GNMax` aggregator, and makes it possible for G-PATE to retain reasonable utility even on high dimensional data. Moreover, random projection preserves similar squared Euclidean distance between high-dimensional vectors, therefore is beneficial to privacy protection both theoretically and empirically [36].

## 5 Privacy Guarantees

In this section, we provide theoretical guarantees on the privacy properties for G-PATE. To start with, we propose the following definition for a differentially private data generative model.

**Definition 3** (Differentially Private Generative Model). Let $G$ be a generative model that maps a set of points $Z$ in the noise space $\mathcal{Z}$ to a set of records $X$ in the data space $\mathcal{X}$. Let $\mathcal{D}$ be the training dataset of $G$ and $\mathcal{A} : \mathcal{D} \mapsto G$ be the training algorithm. We say that $G$ is a $(\varepsilon, \delta)$-*differentially private data generative model* if the training algorithm $\mathcal{A}$ is $(\varepsilon, \delta)$-differentially private.

Definition 3 relaxes the definition of a DP-GAN by focusing the protection only on the generative model in a GAN. This relaxation saves privacy budget during training and improves the utility of the model. Moreover, the relaxation does not compromise the privacy guarantee for the synthetic data.

**Lemma 1.** *Let $G$ be an $(\varepsilon, \delta)$-differentially private data generative model trained on a private dataset $\mathcal{D}$. For any $Z \in \mathcal{Z}$, the synthetic dataset $X = G(Z)$ is $(\varepsilon, \delta)$-differentially private.*

**Proof Sketch.** Lemma 1 is a consequence of the post-processing property of differential privacy. First, the random points $Z$ are independent of the private dataset $\mathcal{D}$. Second, one does not need to query the discriminator during the data generation process. Therefore, the synthetic dataset is generated by post-processing $G$ and is guaranteed to be $(\varepsilon, \delta)$-differentially private.

Lemma 1 shows that a differentially private generative model is able to support infinite number of queries to the data generator and can be used to generate multiple synthetic datasets.

Next, we justify the privacy guarantee of the G-PATE method. The following lemma justifies RDP of the gradient aggregator (Algorithm 2).

**Lemma 2** (Rényi Differential Privacy of `DPGradAgg`). *The output of the gradient aggregator (`DPGradAgg`) proposed in Algorithm 2 satisfies $\left(\lambda, \sum_{1 \leq j \leq k} \varepsilon_j\right)$-RDP, where $\lambda > 1$ and $\varepsilon_j$ is the data-dependent RDP budget with order $\lambda$ for the `Confident-GNMax` aggregator on the $j$-th projected dimension.*

**Proof Sketch.** Lemma 2 can be proved by combining the RDP guarantee of the `Confident-GNMax` aggregator and the post-processing property of RDP. We first divide the input of `DPGradAgg` into two categories. The first category contains data independent parameters, including the gradient clipping constant $c$, the number of bins $B$, the projected dimension $k$, the noise parameters $\sigma_1$ and $\sigma_2$, and the threshold $T$. These parameters do not contain private information. The second category contains the gradient vectors $\Delta \mathbf{X} = (\Delta \mathbf{x^{(1)}}, \ldots, \Delta \mathbf{x^{(n)}})$, which are data-dependent and sensitive. Our privacy analysis focuses on the computation on $\Delta \mathbf{X}$. With random projection and gradient discretization, we convert $\Delta \mathbf{X}$ into $k$ histograms and pass the histograms into the `Confident-GNMax` aggregator. Since `Confident-GNMax` satisfies data-dependent RDP [29], the privacy guarantee nicely propagates to the output of `DPGradAgg`.

We analyze the RDP guarantee of `DPGradAgg` by composing the privacy budget consumed by the `Confident-GNMax` aggregator on each projection dimension. Therefore, the Rényi differential privacy budget of the training algorithm is a composition of the data-dependent Rényi differential privacy budget of the `Confident-GNMax` aggregator over $k$ dimensions. The data-dependent privacy

budget for each `Confident-GNMax` aggregation is dependent on $\sigma_1$, $\sigma_2$, and threshold $T$ (Appendix D). The remaining parameters (e.g. gradient clipping constant $c$, number of bins $B$) do not influence the privacy guarantee.

The next theorem justifies RDP of the G-PATE training process.

**Lemma 3** (Rényi Differential Privacy of G-PATE). *Let $\mathcal{A}$ be the training algorithm for the student generator (Algorithm 1) with $N$ training iterations and $k$ projected dimensions. The data-dependent Rényi differential privacy for $\mathcal{A}$ with order $\lambda > 1$ is $\varepsilon = \sum_{1 \le i \le N} \left( \sum_{1 \le j \le k} \varepsilon_{i,j} \right)$, where $\varepsilon_{i,j}$ is the data-dependent Rényi differential privacy for the* `Confident-GNMax` *aggregator in the $i$-th iteration on the $j$-th projected dimension.*

**Proof Sketch.** For the convenience of privacy analysis, we divide the each iteration in Algorithm 1 into three phases: pre-processing, private computation and aggregation, and post-processing. In the pre-processing phase, the generator produces fake samples without accessing the private data. In the private computation and aggregation phase, the teacher discriminators are updated based on private data. Each teacher discriminator also generates a gradient vector. These vectors are aggregated using the `DPGradAgg` algorithm. Based on Lemma 2, the data-dependent RDP for this phase is $\sum_{1 \le j \le k} \varepsilon_{i,j}$. In the post-processing phase, the student generator is updated using the privately-aggregated gradient vector $\Delta \mathbf{x_j}^{\mathrm{priv}}$. It satisfies RDP because of the post-processing property. Finally, the RDP of Algorithm 1 is composed over $N$ training iterations.

The next theorem provides a theoretical guarantee on the differential privacy of G-PATE.

**Theorem 4** (Differential Privacy of G-PATE). *Given a sensitive dataset $\mathcal{D}$ and parameters $0 < \delta < 1$, let $G$ be the student generator trained by Algorithm 1. There exists $\varepsilon > 0$ and $\lambda > 1$ so that $G$ is a $(\varepsilon + \frac{\log 1/\delta}{\lambda - 1}, \delta)$-differentially private data generative model.*

Theorem 4 is the consequence of converting the Rényi differential privacy guarantee in Lemma 3 to differential privacy (Theorem 1).

# 6 Experimental Evaluation

We evaluate G-PATE against three state-of-the-art models: DP-GAN, PATE-GAN and GS-WGAN. We first perform comparative analysis with baselines on the tabular and image datasets used in the corresponding works, including the Kaggle credit tabular dataset and the grayscale image datasets (MNIST and Fashion-MNIST). In addition, we evaluate G-PATE on the privacy-sensitive large-scale high-dimensional face dataset CelebA.

## 6.1 Experimental Setup

**Tabular Dataset.** we use the same Kaggle credit card fraud detection dataset [7] (Kaggle Credit) as in [37]. The dataset contains 284,807 samples representing transactions made by European cardholders' credit cards in September 2013, and 492 (0.2%) of these samples are fraudulent transactions. Each sample consists of 29 continuous features from a PCA transformation on the original features.

**Image Datasets.** To demonstrate the superiority of G-PATE to PATE-GAN on high dimensional image datasets, we train G-PATE on MNIST, Fashion-MNIST [34], and the celebrity face datasets CelebA [24]. MNIST and Fashion-MNIST consist of 60,000 training examples and 10,000 testing examples. Each example is a $28 \times 28$ grayscale image, associated with a label from 10 classes. The CelebA dataset contains 202,599 images aligned and cropped based on the human face. We create three datasets: CelebA-Gender(S) is a binary classification dataset that uses the gender attributes as the labels and resizes the images to $32 \times 32 \times 3$; to demonstrate the scalability of G-PATE we also create CelebA-Gender(L) with the same label while resizing the images to $64 \times 64 \times 3$; CelebA-Hair contains images as 64x64x3 with three hair color attributes (black/blonde/brown). We follow the official training and testing partition as [24].

**Implementation Details.** For the Kaggle Credit dataset, both the generator and discriminator networks of G-PATE are fully connected neural network with the same architecture as PATE-GAN [37]. We use random projection with 5 projection dimensions during gradient aggregation. We use the DCGAN [32] architecture on the image datasets. We set the projection dimensions to 10 during gradient aggregation. More details are provided in Appendix E.

Table 1: **Performance Comparison on the Tabular Dataset and Image Datasets.** We compare G-PATE with DP-GAN, PATE-GAN, GS-WGAN and vanilla DC-GAN. Vanilla DC-GAN has no privacy protection. The best results are highlighted in bold. Table (a) presents AUROC of the classifier trained on synthetic data and tested on real data. The performance satisfying $(1, 10^{-5})$-differential privacy is evaluated over 4 different classifiers: logistic regression (LR), AdaBoost, bagging, and multi-layer perceptron (MLP).

(a) **Data Utility (AUROC) on Kaggle Credit Tabular Dataset.**

|  | DC-GAN | PATE-GAN | DP-GAN | G-PATE |
|---|---|---|---|---|
| LR | 0.9430 | 0.8728 | 0.8720 | **0.9251** |
| AdaBoost | 0.9416 | 0.8959 | 0.8809 | **0.8981** |
| Bagging | 0.9379 | 0.8877 | 0.8657 | **0.8964** |
| MLP | 0.9444 | 0.8925 | 0.8787 | **0.9093** |
| Average | 0.9417 | 0.8872 | 0.8743 | **0.9072** |

(b) **Visual Quality Evaluation on Image Datasets using Inception Score (IS)**.

| Dataset | Real data | $\varepsilon$ | DP-GAN | PATE-GAN | GS-WGAN | G-PATE |
|---|---|---|---|---|---|---|
| MNIST | 9.86 | 1 | 1.00 | 1.19 | 1.00 | **3.60** |
|  |  | 10 | 1.00 | 1.46 | **8.59** | 5.16 |
| Fashion-MNIST | 9.01 | 1 | 1.03 | 1.69 | 1.00 | **3.41** |
|  |  | 10 | 1.05 | 2.35 | **5.87** | 4.33 |
| CelebA | 1.88 | 1 | 1.00 | 1.15 | 1.00 | **1.17** |
|  |  | 10 | 1.00 | 1.16 | 1.00 | **1.37** |

(c) **Data Utility (Accuracy) on Image Datasets.** The table presents the classification accuracy of CNN models trained on the generated data and tested on real data evaluated under two private settings: $\varepsilon = 10$ and $\varepsilon = 1$ given $\delta = 10^{-5}$.

| Dataset | DC-GAN | $\varepsilon$ | DP-GAN | PATE-GAN | GS-WGAN | G-PATE |
|---|---|---|---|---|---|---|
| MNIST | 0.9653 ($\varepsilon = \infty$) | 1 | 0.4036 | 0.4168 | 0.1432 | **0.5880** |
|  |  | 10 | 0.8011 | 0.6667 | 0.8066 | **0.8092** |
| Fashion-MNIST | 0.8032 ($\varepsilon = \infty$) | 1 | 0.1053 | 0.4222 | 0.1661 | **0.5812** |
|  |  | 10 | 0.6098 | 0.6218 | 0.6579 | **0.6934** |
| CelebA-Gender(S) | 0.8002 ($\varepsilon = \infty$) | 1 | 0.5201 | 0.4448 | 0.6293 | **0.7016** |
|  |  | 10 | 0.5409 | 0.5870 | 0.6326 | **0.7072** |
| CelebA-Gender(L) | 0.8149 ($\varepsilon = \infty$) | 1 | 0.5330 | 0.6068 | 0.5901 | **0.6702** |
|  |  | 10 | 0.5211 | 0.6535 | 0.6136 | **0.6897** |
| CelebA-Hair | 0.7678 ($\varepsilon = \infty$) | 1 | 0.3447 | 0.3789 | 0.3375 | **0.4985** |
|  |  | 10 | 0.3920 | 0.3900 | 0.3725 | **0.6217** |

Table 2: **Data Utility (Accuracy) on Image Datasets given Small Privacy Budgets.** G-PATE and baselines are evaluated following the same way as Table 1c given $\delta = 10^{-5}$ and low $\varepsilon \leq 1.0$.

| $\varepsilon$ | MNIST | | | | Fashion-MNIST | | | |
|---|---|---|---|---|---|---|---|---|
|  | DP-GAN | PATE-GAN | GS-WGAN | G-PATE | DP-GAN | PATE-GAN | GS-WGAN | G-PATE |
| 0.2 | 0.1104 | 0.2176 | 0.0972 | **0.2230** | 0.1021 | 0.1605 | 0.1000 | **0.1874** |
| 0.4 | 0.1524 | 0.2399 | 0.1029 | **0.2478** | 0.1302 | 0.2977 | 0.1001 | **0.3020** |
| 0.6 | 0.1022 | 0.3484 | 0.1044 | **0.4184** | 0.0998 | 0.3698 | 0.1144 | **0.4283** |
| 0.8 | 0.3732 | 0.3571 | 0.1170 | **0.5377** | 0.1210 | 0.3659 | 0.1242 | **0.5258** |
| 1.0 | 0.4046 | 0.4168 | 0.1432 | **0.5880** | 0.1053 | 0.4222 | 0.1661 | **0.5812** |

Table 3: **Visualization of Generated Instances by G-PATE.** Row 1 (real image), row 2 ($\varepsilon = 10, \delta = 10^{-5}$) and row 3 ($\varepsilon = 1, \delta = 10^{-5}$) each presents one image from each class (the left 5 columns are MNIST images, and the right 5 columns are Fashion-MNIST images).

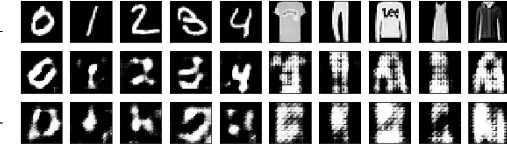

**Evaluation Metrics.** To compare the *data utility* of different data generators, we follow the standard protocol [37, 6] and train a classifier on the synthetic data and test it on the real data to benchmark the usefulness of the synthetic data for downstream tasks. Specifically, we report the **AUROC** of the classifiers for the binary-class tabular dataset, and the **classification accuracy** trained on CNN for the multi-class image datasets to measure the data utility. In addition, for image datasets, we also evaluate the *visual quality* of the synthetic images using **Inception Score (IS)** [18] and **Fréchet inception distance (FID)** [17]. More details can be found in Appendix E and F.

### 6.2 Evaluation Results

**Kaggle Credit.** The Kaggle Credit dataset is highly unbalanced, so we take a two-step approach to generate the unbalanced synthetic data. In the first step, we calculate a differentially private estimation of the class distribution in the training dataset using the Laplacian mechanism [10] with $\varepsilon = 0.01$. In the second step, we train a a $(0.99, 10^{-5})$-differentially private data generator and use it to generate data that follow the estimated class distribution. By the composition theorem of differential privacy [10], the data generation mechanism is $(1, 10^{-5})$-differentially private.

To compare with PATE-GAN, we select 4 commonly used classifiers evaluated in [37] and report the **AUROC** of the 4 classifiers trained on the corresponding synthetic data. We evaluate G-PATE under the same experimental setup as PATE-GAN for $\varepsilon = 1$. The results for baselines are following [37], and we obtain a higher baseline performance for DC-GAN compared to the results reported in [37].

Table 1a presents the data utility analysis based on AUROC between G-PATE and PATE-GAN on Kaggle Credit dataset. G-PATE outperforms both PATE-GAN and DP-GAN and is close to the vanilla DC-GAN which has no privacy protection. The high performance of G-PATE is partly due to the relatively low dimensionality of the Kaggle Credit dataset and the abundance of training examples. More experimental results on Kaggle Credit dataset are presented in Appendix A.

Table 4: **Analysis on the Hyper-parameters.** We performed comprehensive studies on the hyper-parameters of G-PATE (the number of teachers, the projection dimensions, gradient clipping constant $c$, and the number of bins $B$) for MNIST and Fashion-MNIST with $\varepsilon = 1$ and $\delta = 10^{-5}$. "N/A" means "no projections".

| | Projection Dimensions $k$ | | | | # of Teachers $n$ | | | Gradient Clipping Constant $c$ | | | | # of bins $B$ | | |
| | 5 | **10** | 20 | N/A | 2000 | 3000 | **4000** | 5e-4 | 1e-4 | 5e-5 | 1e-5 | 5 | 10 | 20 |
|---|---|---|---|---|---|---|---|---|---|---|---|---|---|---|
| **MNIST** | 0.4638 | **0.5880** | 0.5604 | 0.1141 | 0.4240 | 0.5218 | **0.5880** | 0.4754 | **0.5880** | 0.5505 | 0.4668 | **0.5880** | 0.5810 | 0.4706 |
| **Fashion** | 0.5129 | **0.5812** | 0.5172 | 0.1268 | 0.3997 | 0.4874 | **0.5812** | 0.5140 | 0.5567 | **0.5812** | 0.5339 | 0.5575 | **0.5812** | 0.5400 |

**Image Datsets.** To understand G-PATE's performance on image datasets, we evaluate the *data utility* and *visual quality* of the generated images with G-PATE, PATE-GAN, DP-GAN, and GS-WGAN on the MNIST, Fashion-MNIST, and CelebA datasets. The analysis is performed under two private settings: $\varepsilon = 1, \delta = 10^{-5}$ and $\varepsilon = 10, \delta = 10^{-5}$.

For the data utility, We report the **classification accuracy** of the synthetic data in Table 1c. G-PATE outperforms baselines under both settings, and there is a more significant improvement for the setting with a stronger privacy guarante (i.e., $\varepsilon = 1$). Specifically, we observe that on the Fashion-MNIST dataset the synthetic records generated by DP-GAN under this setting are close to random noise, while the model trained on G-PATE generated data retains an accuracy of 58.12%.

To demonstrate the scalability of our algorithm, we also conduct experiments on the high-dimensional face dateset CelebA. The synthetic data generated by G-PATE is highly utility-preserving, while DP-GAN can barely converge given the high-dimensionality of the data. Particularly, even with the strict privacy budget $\varepsilon = 1$, the accuracy of the generated data by G-PATE on CelebA-Gender(S) is only around 10% lower than the vanilla DC-GAN. Moreover, although the dimensionality of CelebA-Gender(L) is $4\times$ larger than CelebA-Gender(S), the accuracy of both datasets is very close, which again demonstrates the scalability of G-PATE.

To better understand different generative models, we evaluate the visual quality of the generated image data, though it is not the main focus of the differentially private data generator. In particular, we visualize the generated DP images in Table 3. We also provide an quantitative analysis based on **Inception Score** for G-PATE and baselines in Table 1b. G-PATE can consistently generate better images than baselines when $\varepsilon = 1$, for which GS-WGAN does not converge, which demonstrates the superiority of G-PATE. When $\varepsilon = 10$, G-PATE achieves the best performance on the high-dimensional face dataset CelebA. Although GS-WGAN has better visual quality on MNIST and Fashion-MNIST when $\varepsilon = 10$, G-PATE has the best data utility across different settings and datasets, which suggests that the data utility and visual quality are two orthogonal metrics, and it would be an interesting future direction to improve the visual quality. More evaluation details and evaluation of **Fréchet inception distance (FID)** can be found in Appendix F.

**Analysis on the Hyper-parameters.** We perform comprehensive ablation studies on the number of teachers and the the projection dimensions to gain better understanding about G-PATE. As shown in Table 4, G-PATE benefits from having more teacher discriminators. Under the same privacy guarantee, the number of noisy votes ($\sigma_1$ and $\sigma_2$) remains the same, so the output of the noisy voting algorithm is more likely to be correct, and the model would get better performance. However, this benefit diminishes as the training set for each teacher model gets smaller with the increasing number of teachers, and 4000 teachers have already achieved satisfiable results. Table 4 also demonstrates the effectiveness of the random projection method, which improves the classification accuracy by around 47%. With larger projection dimensions, the privacy consumption increases rapidly as we are accessing more private data. But if the projection dimension is too small, useful information can be lost during projection. We find the best trade-off when projection dimension equals to 10. G-PATE achieves better performance given samller bins ($\leq 10$). With larger bins, the teachers attain a lower agreement rate, leading to worse performance.

**Analysis under Limited Privacy Budgets.** We conduct another set of ablation studies given limited privacy budgets. From Table 2 and Figure 2, we can observe that G-PATE starts to converge even under small $\varepsilon$ on both MNIST and Fashion-MNIST datasets. The utility stably increases when the privacy budgets increase. Among different small $\varepsilon$, G-PATE achieves significantly higher accuracy than baselines. In particular, the accuracy of G-PATE under $\varepsilon = 0.6$ is four times higher than DP-GAN, which indicates that G-PATE is able to generate differentially private data with high utility under low privacy budgets.

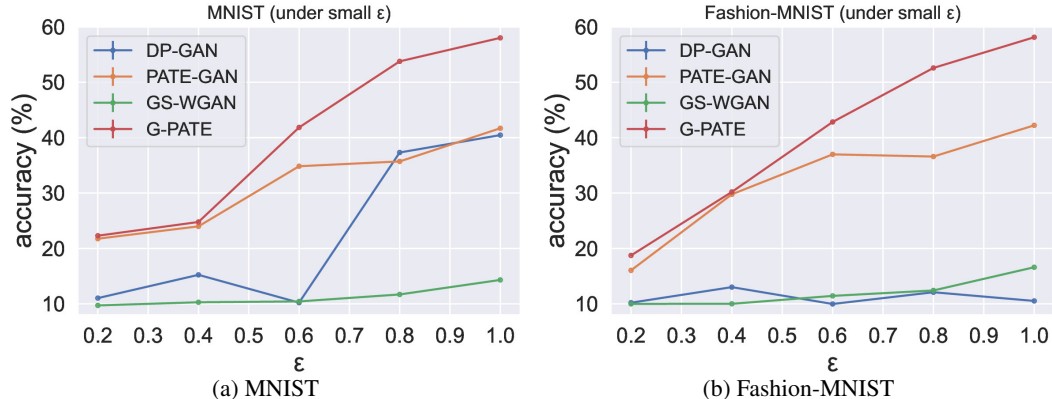

(a) MNIST                                         (b) Fashion-MNIST

Figure 2: **Data Utility (Accuracy) on Image Datasets given Small Privacy Budgets.** The accuracy of G-PATE and baseline models is plotted under tight privacy budget $\delta = 10^{-5}$, $\varepsilon \leq 1.0$. G-PATE consistently outperforms the baseline models even under limited privacy budgets.

**Agreement of Teachers.** In the gradient aggregation step, G-PATE relies on the agreement of teacher models to select the gradient direction. When there is a high agreement rate among teacher models, the gradient aggregator is more robust to noise and more likely to select a gradient direction that preserves higher utility. Intuitively, because the training partitions of teacher models come from the same dataset, we expect the teacher models to learn similar real data distribution. As a consequence, the gradients generated by the teacher models are expected to be similar. Empirically, we evaluate the agreement rate of teacher models on MNIST with 4000 teacher models and $\varepsilon = 1$. On average, the gradient aggregator achieves 60.35% agreement rate for teacher votes on the most agreed direction, and 39.11% on the second agreed direction, which suggests that teacher models have high agreement rates on the top agreement directions.

**Evaluation under a Data-Independent Privacy Budget.** We evaluated the performance of G-PATE on MNIST with a data-independent privacy analysis. We followed the same evaluation process in Table 1c and replaced the privacy analysis with the data-independent privacy bound [29]. When $\varepsilon = 1$, the utility (i.e., classification accuracy) of G-PATE on MNIST is 0.5483, outperforming the existing baselines by a large margin, which demonstrates that G-PATE still achieves the highest data utility with data-independent privacy cost.

**Ablation Study on the Use of PATE.** To understand how the PATE framework contributes to the advantage of G-PATE, we trained a DP-GAN model with gradient discretization and random projection applied to DPSGD on the MNIST dataset under $\varepsilon = 1$ and $\varepsilon = 10$. The model achieves the classification accuracy of 0.2026 ($\varepsilon = 1$) and 0.5602 ($\varepsilon = 10$) respectively. The results demonstrate that the PATE framework contributes significantly to G-PATE's utility advantage. First, with the PATE framework, G-PATE only needs to add noise to one layer of projected gradients between the teacher discriminators and the student generator, so the dimension of the noise equals the data dimension after projection. On the contrary, DP-GAN adds noise to all the gradients of the model, so the dimension of the noise equals to the model dimension. Since the data dimension is usually significantly lower than the model dimension, the PATE framework helps G-PATE to reduce the amount of noise needed to achieve the same privacy guarantee, and therefore preserves better utility. Second, in G-PATE, we have 4000 teachers to vote over the projected gradients and choose the most agreed gradient direction to update the model, which eliminates the noise from the random projection, saves privacy cost, and ensures high utility of the gradients due to the high consensus of teacher discriminators. In comparison, DP-GAN does not have teacher models, and thus the quantized gradients with random projection can contain a lot of noise during aggregation, yielding worse performance.

## 7   Conclusion

We propose G-PATE, a novel approach for training a differentially private data generator for high-dimensional data. G-PATE is enabled by a novel differentially private gradient aggregation mechanism combined with random projection. It significantly outperforms prior work on both image and non-image datasets in terms of preserving data utility for generated datasets. Beyond the high utility compared with the state-of-the-art differentially private data generative models under similar setting, G-PATE is also able to preserve high data utility even given small privacy budgets.

## Acknowledgement

This work was performed under the auspices of the U.S. Department of Energy by the Lawrence Livermore National Laboratory under Contract No. DE-AC52-07NA27344 and LLNL LDRD Program Project No. 20-ER-014 (LLNL-CONF-805494), the NSF grant No.1910100, NSF CNS 20-46726 CAR, and the Amazon Research Award.

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
