# Appendix

## A    Additional Evaluation Results on Kaggle Credit Dataset

In addition to AUROC, we also evaluate the AUPRC of the classification models trained on the synthetic data produced by different generative models. Table 5 presents the results. G-PATE has the best performance among all the differentially private generative models.

|  | GAN | PATE-GAN | DP-GAN | G-PATE |
|---|---|---|---|---|
| **Logistic Regression** | 0.4069 | 0.3907 | 0.3923 | **0.4476** |
| **AdaBoost** | 0.4530 | 0.4366 | 0.4234 | **0.4481** |
| **Bagging** | 0.3303 | 0.3221 | 0.3073 | **0.3503** |
| **Multi-Layer Perceptron** | 0.4790 | 0.4693 | 0.4600 | **0.5109** |
| **Average** | 0.4173 | 0.4046 | 0.3958 | **0.4392** |

Table 5: **AUPRC on Kaggle Credit Dataset.** The table presents AUPRC of classification models trained on synthetic data and tested on real data. PATE-GAN, DP-GAN, and G-PATE all satisfy $(1, 10^{-5})$-differential privacy. The best results among different DP generative models are bolded.

To understand the upper-bound of the classification models' performance. We train the same classification models on real data and test it on real data. The results are presented in Table 6.

|  | LR | AdaBoost | Bagging | MLP |
|---|---|---|---|---|
| **AUROC** | 0.9330 | 0.9802 | 0.9699 | 0.9754 |
| **AUPRC** | 0.6184 | 0.7103 | 0.6707 | 0.8223 |

Table 6: **Performance of Classification Models Trained on Real Data.** The table presents AUROC and AUPRC of classification models trained and tested on real data. These results are the upper-bounds for evaluation results on Kaggle Credit dataset.

## B    Synthetic Images Generated by G-PATE

Figure 3 presents the synthetic images generated by G-PATE on MNIST and Fashion-MNIST. Images in the same column share the same class label. Row 1 contain real images in the training dataset; row 2 contain images generated by G-PATE when $\epsilon = 10, \delta = 10^{-5}$; and row 3 contain images generated by G-PATE when $\epsilon = 1, \delta = 10^{-5}$.

## C    Performance Analysis on Nonprivate GPATE

To understand how the GPATE training framework influence the performance of a GAN, we train a nonprivate GPATE with 10 teacher models. As shown in Table 7, the GPATE structure has a comparable performance to the vanilla GAN.

## D    Privacy Budget of Confident-GNMax

The `Confident-GNMax` aggregator was proposed by [29] to support differentially private aggregation of the votes from multiple teacher models. For the completeness of this paper, in this section, we include the algorithm for the `Confident-GNMax` aggregator and its data-dependent RDP guarantee.

### D.1    The Confident-GNMax Aggregator

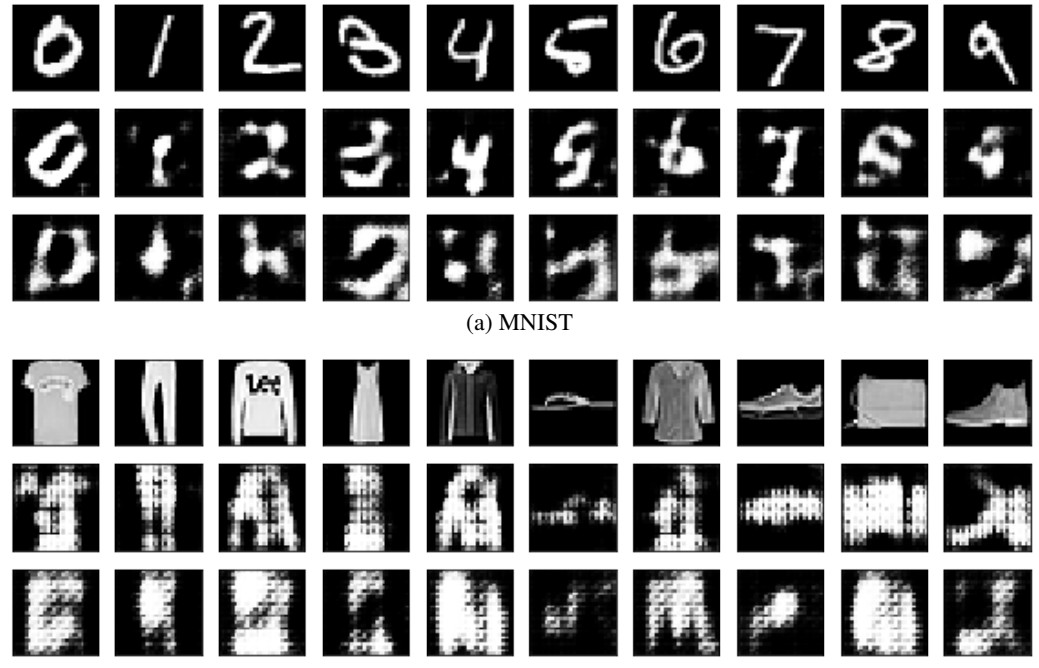

(a) MNIST

(b) Fashion-MNIST

Figure 3: **Visualization of generated instances by G-PATE.** Row 1 (real image), row 2 ($\varepsilon = 10, \delta = 10^{-5}$) and row 3 ($\varepsilon = 1, \delta = 10^{-5}$) each presents one image from each class.

Table 7: **Performance Comparison between GAN and nonprivate GPATE on Kaggle Credit Dataset.**

|  | GAN | Nonprivate GPATE |
|---|---|---|
| **Logistic Regression** | 0.9430 | 0.9455 |
| **AdaBoost [11]** | 0.9416 | 0.9165 |
| **Bagging [4]** | 0.9379 | 0.9456 |
| **Multi-layer Perceptron** | 0.9444 | 0.9219 |
| **Average** | 0.9417 | 0.9324 |

**Algorithm 3 Confident-GNMax Aggregator.** The private aggregator used in the scalable PATE framework [29].

---

**Require:** input $x$, threshold $T$, noise parameters $\sigma_1$ and $\sigma_2$
1: **if** $\max_i\{n_j(x)\} + \mathcal{N}(0, \sigma_1^2) \geq T$ **then**
2:    **Return:** $\arg\max\{n_j(x) + \mathcal{N}(0, \sigma_2^2)\}$
3: **else**
4:    **Return:** $\bot$
5: **end if**

---

Algorithm 3 presents the `Confident-GNMax` aggregator proposed by [29]. The algorithm contains two steps. First, it computes the noisy maximum vote

$$M_1 = \max_i\{n_j(x)\} + \mathcal{N}(0, \sigma_1^2).$$

Then, if the noisy maximum vote is greater than a given threshold, it uses the GNMax mechanism to select the output with most votes:

$$M_2 = \arg\max\{n_j(x) + \mathcal{N}(0, \sigma_2^2)\}.$$

Since each teacher model may cause the maximum number of vote to change at most by 1, $M_1$ is equivalent to a Gaussian mechanism with sensitivity 1. Therefore, following Theorem 3, $M_1$ with Gaussian noise of variance $\sigma_1^2$ guarantees $(\lambda, \lambda/2\sigma_1^2)$-RDP for all $\lambda > 1$.

$M_2$ could be decomposed into post-processing a noisy histogram with Gaussian noise added to each dimension. Since each teacher model may increase the count in one bin and decrease the count in another, the mechanism has a sensitivity of 2. Therefore, $M_2$ with Gaussian noise of variance $\sigma_2^2$ guarantees $(\lambda, \lambda/\sigma_2^2)$-RDP [29].

The data-dependent privacy guarantee for the GNMax mechanism $M_2$ has been analyzed by [29]:

**Theorem 5.** *Let $M$ be a randomized algorithm with $(\mu_1, \varepsilon_1)-RDP$ and $(\mu_2, \varepsilon_2)-RDP$ guarantees and suppose that there exists a likely outcome $i^*$ given a dataset $D$ and a bound $\tilde{q} \leq 1$ such that $\tilde{q} \geq \Pr[\mathcal{M}(D) \neq i^*]$. Additionally, suppose that $\lambda \leq \mu_1$ and $\tilde{q} \leq e^{(\mu_2-1)\varepsilon_2} / \left( \frac{\mu_1}{\mu_1-1} \cdot \frac{\mu_2}{\mu_2-1} \right)^{\mu_2}$. Then, for any neighboring dataset $D'$ of $D$, we have:*

$$D_\lambda \left( \mathcal{M}(D) \| \mathcal{M}(D') \right) \leq \frac{1}{\lambda - 1} \log \left( (1 - \tilde{q}) \cdot \boldsymbol{A} \left( \tilde{q}, \mu_2, \varepsilon_2 \right)^{\lambda-1} + \tilde{q} \cdot \boldsymbol{B} \left( \tilde{q}, \mu_1, \varepsilon_1 \right)^{\lambda-1} \right),$$

*where $\boldsymbol{A} \left( \tilde{q}, \mu_2, \varepsilon_2 \right) \triangleq (1 - \tilde{q}) / \left( 1 - (\tilde{q}e^{\varepsilon_2})^{\frac{\mu_2-1}{\mu_2}} \right)$ and $\boldsymbol{B} \left( \tilde{q}, \mu_1, \varepsilon_1 \right) \triangleq e^{\varepsilon_1} / \tilde{q}^{\frac{1}{\mu_1-1}}$.*

The parameters $\mu_1$ and $\mu_2$ are optimized to get a data-dependent RDP guarantee for any order $\lambda$. By applying Theorem 5 on $M_2$, we obtain the data-dependent RDP budget for $M_2$.

For any $\lambda > 1$, suppose $\varepsilon_1$ is the RDP budget for $M_1$ and $\varepsilon_2$ is the data-dependent RDP budget for $M_2$. Then, the RDP budget for the `Confident-GNMax` algorithm could be calculated as follows:

$$\varepsilon = \begin{cases} \varepsilon_1 & \text{if output is } \perp, \\ \varepsilon_1 + \varepsilon_2 & \text{otherwise.} \end{cases}$$

# E   Model Structures and Hyperparmeters

All of our experiments are running on one AWS GPU server (G4dn.metal) with 8 NVIDIA Tesla T4 GPUs.

**G-PATE.**   For MNIST and Fashion-MNIST, the student generator consists of a fully connected layer with 1024 units and a deconvolutional layer with 64 kernels of size $5 \times 5$ (strides $2 \times 2$). Each teacher discriminator has a convolutional layer with 32 kernels of size $5 \times 5$ (strides $2 \times 2$) and a fully connected layer with 256 units. All layers are concated with the one-hot encoded class label. We apply batch normalization and `Leaky ReLU` on all layers. When $\varepsilon = 10$, we train 2000 teacher discriminators with batch size of 30 and set $\sigma_1 = 600, \sigma_2 = 100$. When $\varepsilon = 1$, we train 4000 teacher discriminators with batch size of 15 and set $\sigma_1 = 3000, \sigma_2 = 1000$. For Kaggle Credit dataset, we train 2100 teacher discriminators with batch size of 32 and set $\sigma_1 = 1500, \sigma_2 = 600$. For all three datasets, we use Adam optimizer [19] with learning rate of $10^{-3}$ to train the models and clip the adversarial perturbations between $\pm 10^{-4}$. The consensus threshold $T$ is set to 0.5.

**GAN.**   The structure of GAN is the same as the structure of G-PATE with a single teacher discriminator. The hyper-parameters are also the same as G-PATE.

**DP-GAN.**   We use DP-GAN method mentioned in [35] on both MNIST and FashionMNIST tasks. For the generator, we use FC Net structure with [128, 256, 512, 784] neurons in each layer, and the discriminator contains [784, 64, 64, 1] neurons in each layer. In each training epoch, the discriminator trains 5 steps and the generator trains 1 step. For both networks, $0.5 \times \text{ReLU}(\cdot)$ activation layers are used. Our batch size is 64 for each sampling, and sampling rate $q$ equals to $\frac{64}{6 \times 10^4}$. We bound the discriminator's parameter weights to $[-0.1, 0.1]$ and kept feature's value between $[-0.5, 0.5]$ during the forward process. In order to generate specific digit data, we concat one-hot vector, which represents digits categories, into each layer in both the discriminator and the generator.

Table 8: **Quality evaluation of images generated by different differentially private data generative models on Image Datasets:** Inception Score (IS) and Frechet Inception Distance (FID) are calculated to measure the visual quality of the generated data under different $\varepsilon$ ($\delta = 10^{-5}$).

(a) $\varepsilon = 1$

| Dataset | Metrics | Real data | DP-GAN | PATE-GAN | GS-WGAN | G-PATE |
|---|---|---|---|---|---|---|
| **MNIST** | IS $\uparrow$ | 9.86 | 1.00 | 1.19 | 1.00 | **3.60** |
| | FID $\downarrow$ | 1.04 | 470.20 | 231.54 | 489.75 | **153.38** |
| **Fashion-MNIST** | IS $\uparrow$ | 9.01 | 1.03 | 1.69 | 1.00 | **3.41** |
| | FID $\downarrow$ | 1.54 | 472.03 | 253.19 | 587.31 | **214.78** |
| **CelebA** | IS $\uparrow$ | 1.88 | 1.00 | 1.15 | 1.00 | **1.17** |
| | FID $\downarrow$ | 2.38 | 485.92 | 434.47 | 437.33 | **293.24** |

(b) $\varepsilon = 10$

| Dataset | Metrics | Real data | DP-GAN | PATE-GAN | GS-WGAN | G-PATE |
|---|---|---|---|---|---|---|
| **MNIST** | IS $\uparrow$ | 9.86 | 1.00 | 1.46 | **8.59** | 5.16 |
| | FID $\downarrow$ | 1.04 | 304.86 | 253.55 | **58.77** | 150.62 |
| **Fashion-MNIST** | IS $\uparrow$ | 9.01 | 1.05 | 2.35 | **5.87** | 4.33 |
| | FID $\downarrow$ | 1.54 | 433.38 | 229.25 | **135.47** | 171.90 |
| **CelebA** | IS $\uparrow$ | 1.88 | 1.00 | 1.16 | 1.00 | **1.37** |
| | FID $\downarrow$ | 2.38 | 485.41 | 424.60 | 432.58 | **305.92** |

**Other Baselines.** We use the default open-source model architecture implementations and hyperparameters for baselines: GS-WGAN[2] and PATE-GAN[3].

**Classification Models for MNIST, Fashion-MNIST, and CelebA.** For each synthetic dataset, we trian a CNN for the classification task. The model has two convolutional layers with 32 and 64 kernels respectively. We use `ReLU` as the activation function and applies dropout on all layers.

**Classification Models for Kaggle Credit.** We implement 4 predictive models in [37] using sklearn: Logistic Regression (`LogisticRegression`), Adaptive Boosting (`AdaBoostClassifier`), Bootstrap Aggregating (`BaggingClassifier`) and Multi-layer Perceptron (`MLPClassifier`). We use *L1* penalty, `Liblinear` solver and *(350:1)* class weight in Logistic Regression. We use logistic regression as classifier in Adaptive Boosting and Bootstrap Aggregating, setting *L2* penalty, number of models as 200 and 100. For Multi-layer Perceptron, we use `tanh` as the activation of 3 layers with 18 nodes and Adam as the optimizer.

# F   Visual Quality Evaluation

We evaluate both Inception Score and Frechet Inception Distance for G-PATE and baselines over MNIST, Fashion-MNIST and CelebA. We present the evaluation results in Table 8.

In our experiments, we follow GS-WGAN and use the implementation[4] for Inception Score calculation with pretrained classifiers trained on real datasets (with test accuracy equal to 99%, 93%, 97% on MNIST, Fashion-MNIST, and CelebA-Gender).

Similarly, we follow GS-WGAN and use the implementation[5] for FID calculation. A lower FID value indicates a smaller discrepancy between the real and generated samples, which corresponds to a better sample quality and diversity.

---

[2]https://github.com/DingfanChen/GS-WGAN
[3]https://bit.ly/3iZZbnx
[4]https://github.com/ChunyuanLI/MNIST_Inception_Score
[5]https://github.com/google/compare_gan

# G  Running Time Analysis on G-PATE

We record the running time of G-PATE on one Tesla T4 GPU under the best parameters (4000 teachers) on MNIST for $\varepsilon = 1$ for three runs. In one epoch, G-PATE takes on average 213.56 seconds for generating fake samples (pre-processing of Phase I in algorithm1) and update parameters of each teacher discriminator (update teacher discriminator in Phase II). Then G-PATE takes on average 43.18 seconds to perform gradient quantization and aggregation (algorithm 2) and update the generator parameters. Overall, G-PATE requires around 256.74 seconds to run for one epoch on MNIST and reaches the privacy budget of $\varepsilon = 1$ at epoch 232, in total 16.5 hours given one single Tesla T4 GPU. In comparison, DP-GAN and PATE-GAN take around 26-34 hours to converge and GS-WGAN requires hundreds of GPU hours to pretrain one thousand non-private GAN as the warm-up steps.

# H  Proofs

## H.1  Proof of Lemma 1

*Proof.* Since $Z \in \mathcal{Z}$ are random points independent of the training dataset $\mathcal{D}$, generation of the synthetic dataset $X = G(Z)$ is post-processing process on the $(\varepsilon, \delta)$-differentially private data generative model $G$. Therefore, $X$ is $(\varepsilon, \delta)$-differentially private based on the post-processing theorem of differential privacy. □

## H.2  Proof of Lemma 2

*Proof.* According to the privacy guarantee of the `Confident-GNMax` aggregation method (Theorem 5), for each dimension $j$ in Algorithm 2, for any $\lambda > 1$, there exists an $\varepsilon_j > 0$ so that the `Confident-GNMax` aggregation method satisfies $(\lambda, \varepsilon_j)$-RDP. Since Algorithm 2 performs `Confident-GNMax` aggregation over $k$ projected dimensions, the privacy guarantee of Algorithm 2 can be derived from by composing the RDP budget over the $k$ dimensions. Therefore, based on the composition theorem of RDP (Theorem 2), Algorithm 2 satisfies $\left(\lambda, \sum_{1 \leq j \leq k} \varepsilon_j\right)$-RDP. □

## H.3  Proof of Lemma 3

*Proof.* First, We apply Lemma 2 to each training iteration in Algorithm 1. For the convenience of privacy analysis, we divide each training iteration into three phases: pre-processing, private computation and aggregation, and post-processing. Based on Lemma 2, the private computation and aggregation phase is $\left(\lambda, \sum_{1 \leq j \leq k} \varepsilon_{i,j}\right)$-RDP, where $\varepsilon_{i,j}$ is the data-dependent Rényi differential privacy for the `Confident-GNMax` aggregator in the $i$-th iteration on the $j$-th projected dimension. Since the pre-processing and post-processing phases do not access the private training dataset, these steps do not increase the RDP budget. Therefore, each training iteration in Algorithm 1 satisfies $\left(\lambda, \sum_{1 \leq j \leq k} \varepsilon_{i,j}\right)$-RDP.

Next, we compose the RDP budget over $N$ iterations. Based on the composition theorem of RDP (Theorem 2), Algorithm 1 satisfies $\left(\lambda, \sum_{1 \leq i \leq N} \left(\sum_{1 \leq j \leq k} \varepsilon_{i,j}\right)\right)$-RDP. □