# OpenReview forum: "G-PATE: Scalable Differentially Private Data Generator via Private Aggregation of Teacher Discriminators"
_NeurIPS.cc/2021/Conference — NeurIPS 2021 Poster_

### Official Review · Reviewer_ouRc · 2021-07-16

**Rating:** 7
**Confidence:** 4

**Summary:**

This paper presents G-PATE, a scalable version of PATE-GAN, that uses a new private gradient aggregation mechanism to
ensure differential privacy on information that flows from the teacher discriminators to the student generator. Experimental results show that G-PATE can work on both tabular and image data and it also show better performance on these data especially for smaller epsilon values compared to existing work such as DP-GAN, PATE-GAN, and few more models.

**Ethics Review Area:**

["I don’t know"]

**Limitations And Societal Impact:**

It would be better if the authors mention the limitations of the proposed work and list few future directions for the improvement.

**Main Review:**

The idea is simple yet effective. The results looks promising with some limitations. Here are some questions/suggestions:

- There are few recent works on synthetic generative models using private embeddings [1][2]. How G-Pate is compared with such models? The author can discuss this in related work (and/or in experiments if it is possible)

- Is there any reference that shows PATE-GAN can only apply on categorical data? Does the authors mean the teacher discriminators output should be categorical or the dataset itself? If the latter, there are works showed PATE-GAN can be used on both categorical and continuous features [3]. Please clarify this in the paper.

- It would be better if the authors showed the results on more tabular data containing both continues and categorical features.

- For Celeb data, the authors resized the images to 32*32 and 64*64. It is interesting to also see how the model performs on high quality images.

- How G-PATE's training time is compared with Pate-GAN and DP-GAN?

- It seems the visualization of generated images are not very good especially for small epsilons. If the quality of generated samples is low what's the advantage of using DP synthesizer vs DP classifiers? Also, how is the quality of generated samples for CelebA data?

[1] https://export.arxiv.org/pdf/2002.11603

[2] https://arxiv.org/abs/2106.04590

[3] https://arxiv.org/pdf/2011.05537.pdf


**Time Spent Reviewing:**

3

---

> ### Author Response · Authors · 2021-08-10
> **Thanks for your valuable comments**
>
> We thank the reviewer for the insightful comments and suggestions, and we provide the responses and additional evaluations below.
>
> > **Q1**: There are few recent works on synthetic generative models using private embeddings [1][2]. How G-Pate is compared with such models? The author can discuss this in related work (and/or in experiments if it is possible)
>
> **A1**:  We thank the reviewer for providing the related work and we will cite and discuss them in our related work. In particular, DP-MERF[1] and PEARL[2] use differentially private embedding to train generative models on the embedding space. Concretely, DP-MERF uses random feature representations of kernel mean embeddings, and PEARL improves upon DP-MERF by incorporating characteristic function that improves the generator’s learning capability. Both DP-MERF and PEARL focus on generating a private embedding space on which the distance metric between synthetic and real data is computed (though there are no results reported for high-dimensional images). On the contrary, G-PATE focuses on generating DP high-dimensional data by improving the model structure and the private gradient aggregation step, which is orthogonal to the embedding space optimization approaches as DP-MERF and PEARL. We believe it would be an interesting future work to integrate these two techniques to optimize both the embedding space and the gradients projection and aggregation to achieve more effective DP generative models.
>
>
> > **Q2**: Is there any reference that shows PATE-GAN can only apply on categorical data? Does the authors mean the teacher discriminators output should be categorical or the dataset itself? If the latter, there are works showed PATE-GAN can be used on both categorical and continuous features [3]. Please clarify this in the paper.
>
> **A2**: Thanks for raising this question! We meant that the teacher discriminators’ output should be categorical, and we will update the paper to be more clear. PATE-GAN can indeed generate both categorical and continuous features, and we have used PATE-GAN to generate image data (which are continuous) as a baseline in our experiments in Table 1(c).
>
> > **Q3**: It would be better if the authors showed the results on more tabular data containing both continues and categorical features.
>
> **A3**: Thanks for the valuable suggestion. We will add more evaluations and update the results soon.
>
> > **Q4**: For Celeb data, the authors resized the images to 3232 and 6464. It is interesting to also see how the model performs on high-quality images.
>
> **A4**: Thanks for the valuable suggestion. We will add more evaluations and update the results soon.
>
> > **Q5**: How G-PATE's training time is compared with Pate-GAN and DP-GAN?
>
> **A5**: Thanks for the valuable comments. We record the running time of G-PATE on one Tesla T4 GPU under the best parameters (4000 teachers) on MNIST for $\varepsilon=1$ for three runs. Overall, G-PATE takes on average 256.74 seconds to run for one epoch on the MNIST dataset and reaches the privacy budget of $\varepsilon=1$ at epoch 232, in total 16.5 hours given one single Tesla T4 GPU. In comparison, DP-GAN and PATE-GAN take around 26-34 hours to converge and GS-WGAN requires hundreds of GPU hours to pretrain one thousand non-private GAN as the warm-up steps. We will add more details in the revision.
>
> > **Q6**: It seems the visualization of generated images are not very good especially for small epsilons.
>
> **A6**: For small $\varepsilon=1$, the quantitative results in terms of Inception Score and Fréchet inception distance in Table 1 (b) and Appendix Table 8 demonstrate that G-PATE can consistently generate consistently higher-quality images than baselines, while state-of-the-art baselines such as GS-WGAN barely converge. We do notice that the generated images are a bit blurry visually given such strong privacy guarantees, and we believe it would be an interesting future direction to improve the visual quality of the generated data under small epsilons which is very challenging.
>
> > **Q7**:  If the quality of generated samples is low what's the advantage of using DP synthesizer vs DP classifiers?
>
> **A7**: The advantage of the DP synthesizer is that the generated DP data is model or downstream task agnostic, and thus it can be applied to various models of different architectures and different sizes and tasks while ensuring differentially privacy. In comparison, DP classifiers can consume a tremendous amount of privacy cost to train large models, as the privacy cost for DP-SGD is proportional to the model size. It is even challenging to train a DP classifier on CIFAR-10 as discussed in [1], not to mention training a DP classifier for high-dimensional data. On the contrary, our DP generative model is able to generate data with DP guarantees and therefore can be used for training different models with privacy guarantees.
>
> [1] ​​Abadi, Martín, Andy Chu, I. Goodfellow, H. B. McMahan, Ilya Mironov, Kunal Talwar and L. Zhang. “Deep Learning with Differential Privacy.” Proceedings of the 2016 ACM SIGSAC Conference on Computer and Communications Security (2016): n. pag.
>
> > **Q8**: Also, how is the quality of generated samples for CelebA data?
>
> **A8**: As shown in Table 1(b) and Table 8, G-PATE can generate higher visual quality data than existing baselines on CelebA under both epsilon=1 and epsilon=10 in terms of Inception Score and Fréchet inception distance.

---

> > ### Author Response · Authors · 2021-08-17
> > **Update Results**
> >
> > > **Q4**: For Celeb data, the authors resized the images to 32x32 and 64x64. It is interesting to also see how the model performs on high-quality images.
> >
> > **A4**: Thanks for the interesting comment. We follow your suggestion and run G-PATE on high-resolution CelebA-Gender. Specifically, since the sizes of CelebA images are different, we normalized the image sizes by resizing the images to 128x128. We observe that the accuracy of G-PATE given epsilon=1 is 0.6270. While we do see some performance drop (from 0.6702 to 0.6270) given the image size change from 64x64 to 128x128, the accuracy is still better than the baselines, and it is even better than the state-of-the-art baseline PATE-GAN of 0.6068 on a lower-resolution dataset CelebA-Gender (L).

---

### Official Review · Reviewer_YNC3 · 2021-07-16

**Rating:** 6
**Confidence:** 4

**Summary:**

This study proposes a new method for generating differentially private synthetic datasets called G-PATE.

This approach combines GANs (Generative Adversarial Neural Networks) with the PATE framework (Private Aggregation of Teacher Ensembles). The proposal utilizes known techniques from the differential privacy literature, such as gradient clipping and discretization.

The new algorithm diverges from previous methods such as DP-GAN, PATE-GAN, and GS-WGAN by creating a differentially private aggregation mechanism for directly using information from teacher discriminators to train the student generator. The method presented in the paper improves upon the previous proposals.

DPGradAgg is designed which aggregates, in a differentially private manner, adversarial perturbations created by each teacher discriminator. The privacy loss of this new method is proven and empirical evaluations on a variety of datasets (Kaggle, MNIST, Fashion-MNIST, CelebA) are shown. The method demonstrates improvements in terms of downstream utility for classification with both smaller and larger epsilon guarantees.

**Limitations And Societal Impact:**

It would be great to see if PATE-G provides improved data utility for subgroups in downstream classification since this is a large challenge currently. This has a societal impact in that there are contentious debates between the impact of DP on the accuracy of analyses for subgroups and any improvements to this would help the adoption of DP. The authors did a good job of listing the limitations of their work especially based on the empirical evaluation at the end of their experimental section.

**Main Review:**

Strong points:
1. The student generator is trained not by ascending the gradient based on the loss from the discriminator but by using the aggregated adversarial perturbations from discriminators.
2. An interesting combination of established techniques, such as gradient clipping, discretization, and random projection, to enable the private release of the gradient at a relatively low privacy cost.

Weak points:
1. The experiments can be improved. For example, authors use the CelebA dataset but not with 40 labels but only with, e.g., the binary labels to determine the gender. Could the authors provide results, for instance, for the CIFAR-10 dataset?
2. The quality of the generated images is low.
3. The proposed idea to incorporate differential privacy into GANs is not novel and improves on existing methods (e.g., PATE-GAN).
4. The method is computationally costly: could authors compare the running times with other methods? What is the execution time of different parts of G-PATE?

Detailed comments:

- Originality:

G-PATE differs from a typical GAN. Specifically, in G-PATE, the student generator is trained not by ascending the gradient based on the loss from the discriminator but by using the aggregated adversarial perturbations from discriminators. The direct comparison between GAN and non-private G-PATE in Appendix C indicates that G-PATE is comparable with GAN but this was done only for the simplest dataset.

This work builds off of PATE-GAN but the method introduces a new mechanism for releasing aggregated gradients (specifically in this case using the aforementioned adversarial perturbations). To my knowledge, this gradient aggregation mechanism is novel and plays a key role in the success of G-PATE over its predecessors such as PATE-GAN & DP-GAN. The study makes it clear how G-PATE differs from other algorithms.

- Quality:

The submission provides both theoretical analyses and empirical evaluations of the proposed method. The theoretical analyses focus on understanding the privacy guarantees of G-PATE. Using the composition and post-processing properties of RDP the authors demonstrate that the privacy loss incurred by the method is simply the composition of the data-dependent RDP from the Confident GNMax aggregator over each component of the random projection dimension. This privacy loss analysis is sound to me. Additionally, the authors use composition over each training iteration to determine the privacy loss of the entire training procedure, which is correct. This analysis demonstrates that their proposed mechanism DPGradAgg is indeed differentially private.

The claims that are made about improved data utility are demonstrated but the results about improved image quality are less conclusive. G-PATE demonstrates improvements across all epsilons and datasets in terms of data utility, but these evaluations can also be improved. Given the high variance in the results of DP learning algorithms, it is unclear whether some of the increases seen would hold over multiple runs. An example of this is MNIST G-PATE vs GS-WGAN at epsilon 10 for measuring the accuracy of a classifier trained on the synthetic data. This is especially true in the data utility results on the Kaggle Credit Tabular dataset between PATE-GAN and G-PATE. When looking at the Inception score results there is no conclusive evidence that G-PATE provides better image quality. The authors address this as a future work consideration in the Inception Score section of the paper at Line 358. A suggestion for the authors to improve the clarity on G-PATE being better than PATE-GAN would be to have graphs that plot these metrics for epsilons between 0.5-10.

- Clarity:

Line 81 is incorrect: "PATE-GAN [33] trains a student generator using an ensemble of teacher discriminators." PATE-GAN trains a student discriminator using an ensemble of teacher discriminators.

Showing the results in Table 1 using graphs over a range of epsilons would help present the improvements over previous methods.

The authors distinguish between their idea and related papers in both the introduction and the related work section. The main contributions of the paper are repeated in a few sections. This should be reduced to single statements without repetitions.

Lines 285 to 290: Theorem 4 is simply a direct copy of Theorem 1. There is no need for proof G.4 in the appendix.

Regarding Table 4, what is the performance of more teachers? Why are there so many more teachers than in the standard PATE?

The lines from 325 to 329 are not entirely clear. The Laplacian Mechanism used there incurs only 0.01 of the privacy budget. The explanation that $\epsilon=0.99$ for a data generator and that data generation mechanism has $\epsilon=1.0$ is relatively a very small difference.

The code is in the supplementary material but it is not prepared to be run. The authors do not specify which versions of the libraries are used (which are rather old versions). For example, which versions of Keras or TensorFlow are used?

Typos:

Line 75 - I believe “us” should be removed? This sentence is not entirely clear.

Line 325: We calculate (not calculates)

Line 374: the privacy consumption increase"S"


- Significance:

This paper can be useful for the community. The empirical results demonstrate good improvements in data utility (which is often the motivation of DP synthetic data over actual image quality).


**Time Spent Reviewing:**

8

---

> ### Author Response · Authors · 2021-08-10
> **Thanks for your valuable comments**
>
> We thank the reviewer for the insightful comments and suggestions, and we provide the responses and additional evaluations below.
>
> > **Q1**: “The experiments can be improved. Could the authors provide results, for instance, for the CIFAR-10 dataset?”
>
> **A1**: Thanks for the suggestion. We aim at the problem of differentially private high-dimensional image generation. Specifically, we conduct experiments on CelebA of size 64x64x3 (larger than CIFAR-10 of 32x32x3), and our G-PATE outperforms existing state-of-the-art baselines by a large margin, which demonstrate the advantages of our proposed gradient quantization and aggregation methods for high-dimensional datasets. Regarding the additional dataset, we follow your suggestion and run G-PATE over another high-dimensional dataset Places365, which consists of 1.8M high-resolution color images of diverse scene categories. We focus on the level-2 classes (Indoor/Outdoor Natural/Outdoor Man-made) and resize the images to 64x64x3. We observe G-PATE can achieve the highest classification accuracy of 34.83 when  $\varepsilon=1, \delta=10^{-5}$, consistently outperforming the existing baselines DP-GAN (0.3200), PATE-GAN (0.3238), GS-WGAN (0.3375). We will add more details on Places365 in the revision.
>
>
> > **Q2**: “The quality of the generated images”
>
> **A2**: We provide the quantitative analysis of visual quality for the generated images in terms of the Inception Score and Fréchet inception distance in Table 1(b) and Table 8. We observe that G-PATE consistently outperforms baselines when  $\varepsilon=1$, which demonstrates that G-PATE can converge faster than baselines. We do observe that when  $\varepsilon=10$, the visual quality is outperformed by the state-of-the-art baseline GS-WGAN. We think the reason is that while our gradient quantization and aggregation approach can help with faster convergence by focusing on the most important gradient information and yield high-utility data, it may also lose some detailed and trivial gradient information for image reconstruction, leading to slightly lower visual quality, but we do want to emphasize that the case where  $\varepsilon=1$ is more important given our privacy protection since it provides much stronger privacy guarantee. We believe it would be an interesting future direction to further improve the visual quality of the generated data when the privacy protection requirement is low.
>
>
> > **Q3**: “The proposed idea to incorporate differential privacy into GANs is not novel and improves on existing methods (e.g., PATE-GAN)”
>
> **A3**: As mentioned in lines 45-58, G-PATE proposed several novel ideas and techniques to support high-dimensional data generation. First, we designed a new model structure to train a student generator directly from an ensemble of teacher discriminators. Second, while PATE-GAN performs the aggregation on the output of a discriminator, our aggregation is performed on the gradients of the discriminator. This change introduces additional challenges in using the PATE framework to aggregate high-dimensional continuous gradient vectors, and we proposed a new gradient aggregation algorithm based on PATE to address this problem. In addition, our proposed random projection and gradient discretization algorithms significantly improve the privacy-utility tradeoff for DP generative models as demonstrated by our extensive experimental results.
>
> > **Q4**: “The method is computationally costly: could authors compare the running times with other methods? What is the execution time of different parts of G-PATE?”
>
> **A4**: Thanks for the valuable comments. We record the running time of G-PATE on one Tesla T4 GPU under the best parameters (4000 teachers) on MNIST for  $\varepsilon=1$ for three runs. In one epoch, G-PATE takes on average 213.56 seconds for generating fake samples (pre-processing of Phase I in algorithm1) and update parameters of each teacher discriminator (update teacher discriminator in Phase II). Then G-PATE takes on average 43.18 seconds to perform gradient quantization and aggregation (algorithm 2) and update the generator parameters.
>
> Overall, G-PATE requires around 256.74 seconds to run for one epoch on MNIST and reaches the privacy budget of  $\varepsilon=1$ at epoch 232, in total 16.5 hours given one single Tesla T4 GPU. In comparison, DP-GAN and PATE-GAN take around 26-34 hours to converge and GS-WGAN requires hundreds of GPU hours to pretrain one thousand non-private GAN as the warm-up steps. We will add these discussions in the revision and thanks for the nice suggestion!
>
> > **Q5**: “A suggestion for the authors to improve the clarity on G-PATE being better than PATE-GAN would be to have graphs that plot these metrics for epsilons between 0.5-10. ”
>
> **A5**: Thanks for the helpful comments. We follow the reviewer’s suggestion and plot the figure of the classification accuracy of different models for small  $\varepsilon$ between 0.2-1, as shown in the figure https://ibb.co/th8CSwR and https://ibb.co/wK070Jp, which again demonstrate the effectiveness of G-PATE.
>
> > **Q6**: “Line 81 is incorrect: "PATE-GAN [33] trains a student generator using an ensemble of teacher discriminators." PATE-GAN trains a student discriminator using an ensemble of teacher discriminators.”
>
> **A6**: Thanks for pointing this out! Indeed, PATE-GAN trains a student discriminator using an ensemble of teacher discriminators, and G-PATE trains a student generator using an ensemble of teacher discriminators. We will update the paper to correct the typo.
>
> > **Q7**: The authors distinguish between their idea and related papers in both the introduction and the related work section. The main contributions of the paper are repeated in a few sections. This should be reduced to single statements without repetitions.
> Lines 285 to 290: Theorem 4 is simply a direct copy of Theorem 1. There is no need for proof G.4 in the appendix.
>
> **A7**: Thanks for pointing this out! We will remove the repeated statements on main contributions in the related work section, and remove the proof in Appendix G.4.
>
>
> > **Q8**: Regarding Table 4, what is the performance of more teachers? Why are there so many more teachers than in the standard PATE?
>
> **A8**: For Table 4, we cannot increase the number of teachers due to the limitation in the size of the MNIST training dataset. Since there are 60000 images in the MNIST training set, we cannot guarantee that each teacher is trained on images from all 10 classes if the number of teachers is greater than 4000.
>
> Increasing the number of teachers would increase the number of votes in the DP aggregation algorithm. As a consequence, the aggregation algorithm would be able to tolerate more noise, which reduces the consumed privacy budget. Compared to the standard PATE, the queries from the students to the teachers in G-PATE are of higher dimension (one for standard PATE), which consumes more privacy budget. Therefore, we need a larger number of teachers to reduce the privacy budget consumed by each query.
>
>
> > **Q9**: The lines from 325 to 329 are not entirely clear. The Laplacian Mechanism used there incurs only 0.01 of the privacy budget. The explanation that  $\varepsilon=0.99$ for a data generator and that data generation mechanism has  $\varepsilon=1.0$ is relatively a very small difference.
>
> **A9**: Thanks for the comments. Lines 325-329 explain the process of synthetic data generation for datasets with an unbalanced class ratio. In the first step, we calculate a differentially private estimation of the class distribution in the training dataset. In the second step, we train a DP generator and use it to generate data that follow the estimated class distribution. Indeed, the first step only incurs a small privacy budget (e.g. 0.01), so its impact on training the DP generator is pretty small. We will update the paper to discuss these two steps more clearly.
>
> > **Q10**: The code is in the supplementary material but it is not prepared to be run. The authors do not specify which versions of the libraries are used (which are rather old versions). For example, which versions of Keras or TensorFlow are used?
>
> **A10**: Thanks for pointing it out. We use TensorFlow 1.15 and Keras 2.24 for the whole experiment. We host our complete package requirements in the requirements.txt here https://www.dropbox.com/s/5yv6ee654epuu6l/requirements.txt?dl=0.

---

> > ### Comment · Reviewer_YNC3 · 2021-08-11
> > **Q1: more classes and Q8: more teachers**
> >
> > Thank you for your answers. I appreciate your effort, especially the new results for the running-time analysis and graphs.
> >
> > Regarding Q1: The first question also should consider more classes. As pointed in the main review, the CelebA dataset was used but not with 40 labels but only with, e.g., the binary labels to determine the gender. Could the authors provide results, for instance, for the CIFAR-10 dataset? How many classes were used for Places365?
> >
> > Regarding Q2: How could you improve your method for the higher privacy budgets, e.g, $\varepsilon=10$.
> >
> > Regarding Q8: there could be 6000 teachers, each teacher with a single image for each digit. This concern regarding the impact of PATE on your method was also raised by the reviewer **Hstn**. Would you show the performance of your method against the number of teachers used?
> >
> > Regarding the code, it is not possible to reproduce the results. You should provide a README.md file that would point to the functions/methods where you do gradient clipping, discretization, random projection, etc.

---

> > > ### Author Response · Authors · 2021-08-12
> > > **Thank you for the valuable follow-up feedback**
> > >
> > > Thank you for the valuable follow-up feedback.
> > >
> > > > **Q1**:  The first question also should consider more classes. CelebA dataset was used but not with 40 labels but only with, e.g., the binary labels to determine the gender.
> > >
> > > **A1**: We are sorry for the confusion. Actually, besides the binary gender labels, we also experimented with three-class hair attributes (Brown/Black/Blonde) on the CelebA-Hair dataset (line 309-310). As shown in Table 1(c), even with this non-binary setting, G-PATE can achieve high accuracy as 0.4985 (epsilon=1) and 0.6217 (epsilon=10), substantially outperforming the state-of-the-art baselines. We will make this clear in our paper.
> > > In our newly added Place 365 dataset results, we also use the three-class classification setting (Indoor/Outdoor Natural/Outdoor Man-made) as mentioned in our previous response.
> > >
> > > In addition, though the CelebA dataset has 40 attributes, these attributes are not mutually exclusive, thus not suitable for training a single-label DP conditional GAN. **To further understand how more classes might impact the results, we can create another CelebA dataset with four-class hair color that adds an additional “other” attribute (Brown/Black/Blonde/Other) and evaluate the performance of G-PATE. Please let us know whether you think this four-class setting is sufficient. If so, we can run these new experiments and provide the results. Thanks!**
> > >
> > >
> > > > **Q2**. How many classes were used for Places365?
> > >
> > > **A2**: We are sorry for the confusion. As discussed above, the Places365 we used consists of three classes (Indoor/Outdoor Natural/Outdoor Man-made). Each class contains 40k images resized to 64x64x3. We will add more details of the Places365 dataset in the revision to make it clear.
> > >
> > > > **Q3**. Regarding Q2: How could you improve your method for the higher privacy budgets, e.g,
> > > ε=10
> > >
> > > **A3**: Thanks for the interesting question. We first note that higher privacy budget (e.g., epsilon=10) scenarios are not very useful in the context of privacy protection, since such privacy guarantees are too loose. Thus, in our paper, we mainly focus on demonstrating the high data utility under scenarios with a strong privacy guarantee (e.g., low epsilon).
> > >
> > > To improve the image quality given large privacy budgets, **there are mainly two folds of directions to follow: 1) Improve the model architectures of the backbone generative model.** For instance, one can adopt a more advanced discriminator (e.g., ResNet) and generator, since GPate is model-agnostic. Currently, we are using the basic convolutional DC-GAN as the backbone, and thus there is some room for improvement given more advanced neural network architectures; **2) More careful parameter tuning.** For instance, one can carefully tune the confidence threshold in the PATE framework, such that we can better save the privacy budget and improve generated data quality. Currently, we select confidence threshold = |0.5 * #/teachers| for MNIST and Fashion-MNIST datasets for simplicity, and more careful tuning would definitely further improve the performance.
> > >
> > > Finally, we want to note that as mentioned in line 362-363, data utility and image visual quality are two orthogonal metrics and our main focus is to improve the data utility for high-dimensional datasets. We do believe that it would be an interesting future work to improve the visual quality for differentially private image generators.
> > >
> > > > **Q4**. There could be 6000 teachers, each teacher with a single image for each digit. Would you show the performance of your method against the number of teachers used?
> > >
> > > **A4**: Thanks for the valuable comment. Since the MNIST training dataset is slightly imbalanced; for instance, the digit “5” only has 5421 samples, we cannot guarantee that each teacher is trained on images from all 10 classes with 6000 teacher discriminators. As shown in Table 4, the performance of G-PATE increases when the number of teachers increases from 2000 to 4000, since theoretically more teachers can tolerate more noise and help reduce the privacy cost of each query. **We can launch an additional experiment to evaluate G-PATE with 5000 teachers on MNIST. Please let us know if you think this experiment is sufficient. If so, we can run this new experiment and provide the results. Thanks!**
> > >
> > >
> > > > **Q5**: This concern regarding the impact of PATE on your method was also raised by the reviewer Hstn.
> > >
> > > **A5**: Thanks for pointing it out. We want to clarify that the concern raised by reviewer Hstn is “which part of GPate (gradient discretization and aggregation, or PATE mechanism) contributes more to the improvements”. We follow the reviewer’s suggestion and run DP-GAN with our gradient discretization and aggregation method, and the results show low data utility, which indicates that in addition to the gradient discretization and aggregation the PATE mechanism also contributes to the final performance as expected. We will add this interesting discussion to our revision.
> > >
> > >
> > > > **Q6**: Regarding the code, it is not possible to reproduce the results. You should provide a README.md file that would point to the functions/methods where you do gradient clipping, discretization, random projection, etc.
> > >
> > > **A6**: We are sorry for the confusion. We do have a README file with an example command in our supplementary materials. We also provide an additional example command below that will give the results as shown in paper (Table 1(c)) for G-PATE on the MNIST dataset, after the MNIST dataset is downloaded and unzipped in the default `data_dir` directory (`../../data`).
> > >
> > > In particular, the Gradient clipping is implemented in line 176 in function `gradient_voting_rdp` in `rdp_utils.py`. Gradient discretization is implemented in line 178-185 in function `gradient_voting_rdp` in `rdp_utils.py`. Random projection is implemented in line 166-174 in function `gradient_voting_rdp` in `rdp_utils.py`, where the Guassian random projection matrix is initialized in line 334-336 in function `aggregate_results` in `model.py`. We will make it clear in the revision.
> > >
> > > Example of command for running:
> > > ```
> > > python main.py --checkpoint_dir mnist_teacher_4000_z_dim_50_c_1e-4/ --teachers_batch 40 --batch_teachers 100 --dataset mnist --train --sigma_thresh 3000 --sigma 1000 --step_size 1e-4 --max_eps 1 --nopretrain --z_dim 50 --batch_size 64
> > > ```

---

> > > > ### Author Response · Authors · 2021-08-17
> > > > **Update Results**
> > > >
> > > > > **Q4**: There could be 6000 teachers, each teacher with a single image for each digit. Would you show the performance of your method against the number of teachers used?
> > > >
> > > > **A4**: Thanks for the valuable comment. Since the MNIST training dataset is slightly imbalanced; for instance, the digit “5” only has 5421 samples, we cannot guarantee that each teacher is trained on images from all 10 classes with 6000 teacher discriminators. As shown in Table 4, the performance of G-PATE increases when the number of teachers increases from 2000 to 4000, since theoretically more teachers can tolerate more noise and help reduce the privacy cost of each query.
> > > >
> > > > In particular, following the suggestion, we have conducted an additional experiment to evaluate G-PATE with 5000 teachers on MNIST. Please let us know if you think this experiment is sufficient. Thanks!
> > > >
> > > > **Update**: We have launched this additional experiment to evaluate G-PATE with 5000 teachers on MNIST. Given epsilon=1, the accuracy of G-PATE is 0.6229, which improves the best performance of G-PATE on MNIST currently (0.5880) given the rigorous privacy constraint epsilon=1 and outperforms all the baselines substantially. The results are aligned with our analysis and as expected that more teachers can help reduce the privacy cost and improve the data utility.
> > > > Please let us know if you have further suggestions or comments.

---

> > > > > ### Comment · Reviewer_YNC3 · 2021-08-25
> > > > > **Comparison with a standard PATE student model**
> > > > >
> > > > > Thank you for your responses. There was no confusion regarding questions Q1 and Q2 - it was my concern that there was a small number of classes (2 or 3 only).
> > > > >
> > > > > Would you also check how good your generator is? For example, on MNIST, your generator can be used to provide training data points that are used to train a model. This model can be used for prediction and compared against a standard PATE student model that was trained with the same privacy budget. What is the accuracy of the model trained using your generator vs the standard PATE student model?
> > > > >
> > > > > You could use privacy budget $\varepsilon=1.97$, the same as in the 2nd PATE paper [1], where the accuracy of the student model was 98.5% (Table 1).
> > > > >
> > > > > [1] Nicolas Papernot, Shuang Song, Ilya Mironov, Ananth Raghunathan, Kunal Talwar, Úlfar Erlingsson. Scalable Private Learning with PATE. ICLR 2018.

---

> > > > > > ### Author Response · Authors · 2021-08-26
> > > > > > **Thank you for the additional comments and questions**
> > > > > >
> > > > > > Thank you for the additional comments and questions!
> > > > > >
> > > > > > For Q1 & Q2, we’d like to highlight that there are experiment results on datasets with more classes in the paper (e.g. both MNIST and Fashion-MNIST have 10 classes). We used 2 or 3 classes for CelebA because the 40 labels on CelebA are **not** mutually exclusive. Concretely, CelebA contains 40 attributes where each attribute is a binary or 3-class label.
> > > > > >
> > > > > > > **Q**: What is the accuracy of the model trained using your generator vs the standard PATE student model?
> > > > > >
> > > > > > **A**: Thanks for the suggestion, and indeed this is how we evaluate our proposed pipeline in the paper. Concretely, we have evaluated the performance of G-PATE generators following your suggestions: on each evaluated dataset, we used the generators to generate synthetic data and used the synthetic data to train a classifier. We evaluated the data utility using the accuracy of the classifier and compared the utility with **state-of-the-art DP generative models**. The results are presented in Table 1 and Table 2 in the paper. We will try to update section 6 to make the evaluation steps more clear.
> > > > > >
> > > > > > We’d like to clarify that, because the standard PATE model is a DP classification model rather than data generative model, it’s **not** directly comparable to DP generative models here. Concretely, DP classification models are designed for a specific classification task and only predict the **label** of a given image. On the contrary, DP generative models produce differentially private **synthetic data** that can be used for training **any** tasks (e.g., we have trained classifiers using the generated data and verified the utility of the trained classifiers as the reviewer suggested).
> > > > > >
> > > > > > Overall, in this paper, we followed the standard process to evaluate the utility and visual quality of the generated data used in prior work on DP generative models [31, 33, 6].

---

> > > > > > > ### Comment · Reviewer_YNC3 · 2021-08-27
> > > > > > > **Comparison: Utility vs Privacy**
> > > > > > >
> > > > > > > Thank you for the answer. I agree with the authors that the student model in the standard PATE framework is trained for a single task, namely the multiclass classification. The PATE student model does clearly outperform the model trained with images generated from G-PATE. The main point is to see these gaps in terms of utility vs privacy. The below table is for MNIST, where the methods are presented in ascending order with respect to their accuracy:
> > > > > > >
> > > > > > > | method | epsilon  | accuracy (%) |
> > > > > > > |--------|----------|--------------|
> > > > > > > | **G-PATE** | 1        | 58.80        |
> > > > > > > | **G-PATE** | 10       | 80.92        |
> > > > > > > | **DC-GAN** | $\infty$ | 96.53        |
> > > > > > > | **PATE**   | 1.97     | 98.5         |
> > > > > > > | **Non-private** | $\infty$ | 99.9 |
> > > > > > >
> > > > > > > The gap in terms of the utility vs privacy tradeoff between PATE and G-PATE is rather substantial.

---

> > > > > > > > ### Author Response · Authors · 2021-08-27
> > > > > > > > **Utility Comparison between Classification Models and Generative Models**
> > > > > > > >
> > > > > > > > Thank you for the comments!  We agree with the reviewer that the performance of standard PATE outperforms all state-of-art DP generative models along the *single* multi-class classification task PATE is designed for. We’d like to highlight that this difference in utility is not caused by the design of G-PATE or any other DP generative models, but by the different purposes of these two categories models.
> > > > > > > >
> > > > > > > > For instance, the results provided by the reviewer supports this statement well: the accuracy of PATE even outperforms the non-private DC-GAN on the single classification task which is the upper bound of G-PATE and all other DP generative models. We will add these discussions in our revision.
> > > > > > > >
> > > > > > > > That is to say, since classification models are designed for a single task, they can achieve better performance on that specific task by design. Meanwhile, generative models support a broader range of tasks and allow the users to directly access the synthetic data for general purpose. Therefore, the utility of a single-task classification model is not directly comparable to the utility of a data generative model, but it is indeed interesting to show and understand the gaps between these different types of models and we will also add these discussions.
> > > > > > > > Please let us know if you have further questions and suggestions!

---

### Official Review · Reviewer_f9CE · 2021-07-16

**Rating:** 7
**Confidence:** 4

**Summary:**

The paper proposes a new method called G-PATE for generating differentially private datasets. G-PATE is motivated from the PATE framework, by training teacher discriminators on real datasets, and then privately aggregating the gradients from those discriminators to train the generator. To aggregate the gradients, the paper proposes to project the gradients to a lower dimension, discretize the gradients, and then use the voting mechanism from PATE to aggregate the gradients. The paper demonstrates that G-PATE achieves better fidelity-privacy trade-offs than the two prior techniques: DP-GAN and PATE-GAN.

**Limitations And Societal Impact:**

The authors discussed the limitations, but not the potential negative societal impact of the work.

**Main Review:**

Privacy-preserving data generation is an important problem in both the research community and the industry. GANs can generate high-fidelity (non-private) data much better than many alternatives, and therefore, there are many interests in applying GANs for privacy-preserving data generation. However, the two main classes of approaches in this direction (i.e. DP-GAN and PATE-GAN) achieve bad fidelity-privacy trade-offs.

The approaches in this paper are very interesting. The idea of only making the generator differentially private while leaving the discriminator sensitive is a very good innovation, which makes it substantially different from PATE-GAN despite that both are motivated by the PATE framework. The idea of using the voting mechanism from PATE to aggregate the gradients is also very interesting, which utilizes the benefit of having multiple teachers, and makes it substantially different from DP-GAN.

The paper also conducts experiments on many different datasets and evaluation tasks, and demonstrates the superiority over DP-GAN, PATE-GAN, and GS-WGAN.

Therefore, I would regard G-PATE as a completely new class of approaches for solving this problem. I can see that this paper will open up many follow-up works and create a new opportunity on solving this important problem. I think this is a very good paper with solid contributions.

I only have one minor suggestion.
* There is an inconsistency in how you explain your proposed gradient aggregation mechanism. In many parts of the paper (e.g. abstract, introduction), you say you are aggregating the gradient in a privacy-preserving way. However, in section 4.1, you say you are generating perturbations for the generated images, and then solving a mean squared error objective over the generator. From my understanding, both of the two explanations are correct and are indeed equivalent, because the gradient of Eq. 2  is exactly (a scaled version of) \delta x, the backpropagated gradient in the discriminator till the input layer. However, mixing these two explanations up in the paper does create some confusion. I would recommend using a consistent explanation, and mention the other as an alternative understanding of the approach in Section 4.1. But the current presentation is also fine.

**Time Spent Reviewing:**

3 hours

---

> ### Author Response · Authors · 2021-08-10
> **Thanks for your valuable comments**
>
> We thank the reviewer for the helpful comments and suggestions, and we will follow the suggestions to improve our paper. We provide responses to the questions below.
>
> > **Q1**: There is an inconsistency in how you explain your proposed gradient aggregation mechanism.
>
> **A1**: Thanks for the valuable comments! Indeed, the perturbation in section 4.1 is a scaled version of the backpropagated gradient in the discriminator till the input layer. We used the perturbation to explain how we calculated the gradient and added noise to achieve DP. We are sorry that it created confusion. We will update the paper to consistently use the term “gradients''. Specifically, we will update the “generating adversarial perturbation step” in Section 4.1 to “Backpropagating gradients in the discriminator”  and use the adversarial perturbation explanation as an alternative understanding of gradient backpropagation.

---

> > ### Comment · Reviewer_f9CE · 2021-08-17
> > **Thanks for the response**
> >
> > Thank the authors for the response! The answer clears my question. I also read authors' responses to other reviews and decide to keep the score as they do not change my belief in the value of this work.

---

### Official Review · Reviewer_Hstn · 2021-07-17

**Rating:** 6
**Confidence:** 4

**Summary:**

The paper studies the differentially private data generator problem. The authors propose the G-PATE algorithm based on the PATE framework.  The G-PATE trains a student data generator with an ensemble of teacher discriminators.  In their approach, they combine the random projection and gradient discretization to release the aggregated gradient privately.  Empirically, they demonstrate that G-PATE improves the privacy-utility trade-offs over prior work.


**Limitations And Societal Impact:**

The authors did not discuss the potential negative societal impact.

**Main Review:**

Clarity: The paper is generally written well though the authors reiterate their contributions too much.

Question:  In the standard PATE, the confidence parameter T is set to |#teachers*0.7|.  I wonder if the authors use the same confidence threshold T in all gradient positions or T needs to be tuned at different gradient locations?

Weak points:
1. My main concern is that the privacy guarantee seems a little bit artificial to me. The privacy loss of G-PATE scales to both the number of queries and the model dimension after projection. DP-GAN can benefit from privacy amplification by subsampling though its privacy loss is model-dependent. Moreover, PATE and PATE-GAN are model-agnostic, and the privacy cost is proportional to the number of queries while G-PATE is model-dependent. It makes me surprised that the proposed G-PATE take the disadvantages of both PATE and DP-GAN can beat both approaches. I suspect the advantage of the proposed G-PATE comes from the random projection and gradient discretization instead of the PATE framework. It will be interesting to investigate whether a DP-GAN (combined with gradient discretization and projection) is comparable with G-PATE.

2. The privacy cost comparison in the experiments might be unfair. G-PATE reports the privacy cost under data-dependent RDP while other baselines (DP-GAN) report the standard DP. Note that the data-dependent RDP itself is private and is much smaller compared to its data-independent counterpart. I encourage the authors to either report the data-independent privacy cost or use the similar smooth sensitivity from the original PATE to sanitize the privacy cost.

Strong points:
1. The proposed gradient aggregation mechanism is novel and can be incorporated into other differentially private deep learning frameworks.
2. The experimental results looks good to me despite the privacy comparison issues mentioned above.





**Time Spent Reviewing:**

7 hours.

---

> ### Author Response · Authors · 2021-08-10
> **Thanks for your valuable comments**
>
> We thank the reviewer for the insightful comments and suggestions and we provide our responses and additional evaluation below.
>
> > **Q1**: In the standard PATE, the confidence parameter T is set to |#teachers*0.7|. I wonder if the authors use the same confidence threshold T in all gradient positions or T needs to be tuned at different gradient locations?
>
> **A1**: Thanks for the comment. We use the same confidence threshold in all gradient positions given one dataset. T is only  tuned for different datasets: for image datasets of smaller dimension (e.g., MNIST, Fashion-MNIST), we choose T=|#teachers\*0.5|; for image datasets of larger dimension (e.g., CelebA), we choose a larger T=|#teachers\*0.8|, since higher-dimensional image datasets consume much more privacy budgets and a higher confidence parameter T can help save the privacy budget given that the G-PATE will update the models only when more than T teachers are agreed.
>
>
>
> > **Q2**: Concern about the privacy loss of G-PATE scales to both the number of queries and the model dimension after projection.
>
> **A2**: Thanks for the valuable comments. We’d like to point out a major distinction between G-PATE and DP-GAN. Although both algorithms add noise to the gradient of the model, G-PATE only needs to add noise to one layer of projected gradients between the discriminator and the generator, while DP-GAN adds noise to all the gradients of the model. Therefore, the privacy loss of G-PATE scales to the number of training iterations (i.e. the number of queries to the teacher discriminators) and the data dimension after projection rather than model dimension. On the contrary, the privacy loss of DP-GAN scales to both the number of training iterations and model dimensions. Since the data dimension is usually significantly lower than the model dimension, G-PATE achieves a much higher utility than DP-GAN.
>
>
>
> > **Q3**: “The advantage of the proposed G-PATE comes from the random projection and gradient discretization instead of the PATE framework. It will be interesting to investigate whether a DP-GAN (combined with gradient discretization and projection) is comparable with G-PATE.”
>
> **A3**: Thanks for the insightful comments. We follow your suggestion and run DP-GAN with gradient discretization and projection on MNIST under  $\varepsilon=1$ and $10$, achieving the classification accuracy of 0.2026 ($\varepsilon=1$) and 0.5602 ($\varepsilon=10$). We think the reason why the results are low for DP-GAN is that the PATE mechanism is also important for our gradient aggregation approach to save privacy budget.
>
> Concretely, in G-PATE, we have 4000 teachers to vote over the projected gradients and choose the most agreed gradient direction to update the model, which eliminates the noise from the random projection, saves privacy cost, and ensures high utility of the gradients due to the high consensus of teacher discriminators. In comparison, DP-GAN does not have teacher models, and thus the quantized gradients with random projection can contain a lot of noise during aggregation, yielding worse performance.
>
> > **Q4**: “The privacy cost comparison in the experiments might be unfair. Report the data-independent privacy cost”
>
> **A4**:  Thanks for the valuable comments. We follow your suggestion and run G-Pate on MNIST with data-independent privacy cost given $\varepsilon=1$. The classification accuracy is 0.5483, outperforming the existing baselines by a large margin, which demonstrates that G-Pate still achieves the highest data utility with data-independent privacy cost. We will add more results on data-independent privacy loss and discussions in the revision.

---

### Decision · Program_Chairs · 2021-09-27

**Decision:**

Accept (Poster)

**Comment:**

The reviews for the paper were mixed. Unfortunately, the reviewers still did not change the score even after the rebuttal. I am still supporting acceptance because of the novel gradient aggregation scheme in the paper.

In the following I list some of the concerns that may not have been addressed adequately in the rebuttal, and the authors need to take care of them.

1. There was a concern that the advantage of the proposed G-PATE comes from the random projection and gradient discretization instead of the PATE framework. The authors did some initial experiments on DP-GAN to show that that the gradient discretization actually impacts the improvement in accuracy. However, these experiments are fairly preliminary, and needs a thorough discussion in the main text of the paper.

2. The quality of the images generated were not that great.  There were some concerns about the efficacy of the algorithm at higher epsilons >10, when compared to other approaches. Since single digit epsilons seem to be mostly the industry standard as of now, it is important to understand the efficacy of the algorithm in this privacy regime.